



# Pingo development in Grøndalen, West Spitsbergen

Nikita Demidov[1], Sebastian Wetterich[2], Sergey Verkulich[1], Aleksey Ekaykin[1,3], Hanno Meyer[2], Mikhail Anisimov[1,3], Lutz Schirrmeister[2], Vasily Demidov[1], Andrew J. Hodson[4, 5]

[1]Arctic and Antarctic Research Institute, Bering St. 38, 199397 St. Petersburg, Russia

[2]Alfred Wegener Institute Helmholtz Center for Polar and Marine Research, Telegrafenberg A45, D-14473 Potsdam, Germany

[3]St. Petersburg State University, 10[th] Line 33-35, 199178 St. Petersburg, Russia

[4]University Centre in Svalbard, N-9171 Longyearbyen, Norway

[5]Western Norway University of Applied Sciences, Røyrgata 6, N-6856 Sogndal, Norway

*Correspondence to*: Nikita Demidov (nikdemidov@mail.ru)

**Abstract.** Pingos are common features in permafrost regions that form by subsurface massive-ice aggradation and create hill-like landforms. Pingos on Spitsbergen have been previously studied to explore their structure, formation timing, connection to springs as well as their role in post-glacial landform evolution. However, detailed hydrochemical and stable-isotope studies of massive ice samples recovered by drilling has yet to be used to study the origin and freezing conditions in pingos. Our core record of 20.7 m thick massive pingo ice from Grøndalen differentiates into four units: two characterised by decreasing $\delta^{18}O$ and $\delta D$ and increasing $d$ (units I and III), and two others show the opposite trend (units II and IV). These delineate changes between episodes of closed-system freezing with only slight recharge inversions of the water reservoir, and more complicated episodes of groundwater freezing under semi-closed conditions when the reservoir got recharged. The water source for pingo formation shows similarity to spring water data from the valley with prevalent $Na^+$ and $HCO_3^-$ ions. The sub-permafrost groundwater originates from subglacial meltwater that most probably followed the fault structures of Grøndalen and Bøhmdalen. Today the pingo of Grøndalen is relict and degrading due to warming surface temperatures. The state of pingos on Spitsbergen depends on complex interaction of climate, permafrost and groundwater hydrology conditions, and is thus highly sensitive to climate warming.

## 1 Introduction

Pingos are widespread landforms that occur within the permafrost zone of the Earth (Grosse and Jones, 2011) and likely on Mars (Burr et al., 2009). The distribution of pingos is closely linked to permafrost history, underground hydrology and climate conditions. Two pingo types are commonly distinguished, which are (1) hydrostatic (closed) system and (2)



hydraulic (open) system pingos. Hydrostatic pingos form when a distinct volume of pore water in water-saturated deposits expulses towards the freezing front and freezes. Hydraulic pingos occur where pressurised groundwater inflow from within or below permafrost freezes at the freezing front. Both processes result in a massive ice body composed of intrusive and/or segregation ice that heaves the surface and form conical elevations. Pingos have a characteristic elliptical to circular planar

shape reaching diameters of up to several hundred meters and heights of up to several dozen meters.

Pingos are rather well studied in Alaska and Canada in terms of formation (e.g. Mackay, 1962), structure (e.g. Yoshikawa et al., 2006) and distribution (e.g. Jones et al., 2012). Pingo growth and decay rates, pingo age, and past distribution of those landforms have been used for the reconstruction of past periglacial landscape conditions (e.g. Mackay, 1986). Pingo ice and sedimentary inventories were investigated and furthermore employed in palaeoenvironmental reconstructions in the

Mackenzie Delta in Canada (Hyvärinen and Ritchie, 1975), on Seward Peninsula in Alaska (Wetterich et al., 2012; Palagushkina et al., 2017), in Siberia (Ulrich et al. 2017; Chizhova and Vasil'chuk, 2018; Wetterich et al., 2018), where they are called 'bulgunniakhs', and in northern Mongolia (Yoshikawa et al., 2013; Ishikawa and Yamkhin, 2016).

Pingos on Spitsbergen (76 identified pingos by Hjelle, 1993) are commonly attributed to open-system conditions (Liestøl, 1977), and have previously been studied by geophysical techniques (e.g. Ross et al., 2005; Rossi et al., 2018) and combined

chemical and physical investigation of the properties of pingo ice (e.g. Yoshikawa, 1993; Yoshikawa and Harada, 1995). Unlike in other permafrost regions, stable isotope properties of the pingo ice were rarely studied on Spitsbergen (Matsuoka et al., 2004).

Yoshikawa and Harada (1995) proposed three formation mechanisms for open-system pingos on Spitsbergen. They differentiate into pingos fed by sub-permafrost groundwater along geologic faults (group I), pingos fed by artesian flow of

migrating sub-glacial groundwater mainly in river valley positions (*in sensu* Liestøl, 1977, group II) and pingos, which are found in nearshore environments of post-glacial isostatic uplift and fed through small-scale discontinuities 'groundwater dikes' (*in sensu* Yoshikawa and Harada, 1995, group III) or taliks in permafrost that aggrades in marine deposits. Yoshikawa and Harada (1995) regard the pingos of the Grøndalen as ancient group III pingos, which formed in Holocene marine sediments quickly after sea regression. However, as Liestøl (1996) recognised 'In connection with formation of pingos there

are a great many unsolved questions. Drillings and temperature measurements through the pingo mound and also through the surrounding permafrost are needed before the problems can be better understood'. To shed light on the Grøndalen pingo formation in comparison to other pingo records from Spitsbergen, we here present the stable isotope and hydrochemical inventories of an entirely cored pingo in Grøndalen near Barentsburg (West Spitsbergen, Fig. 1). Additional data for consideration of postglacial landform evolution in the Grøndalen were obtained from neighbouring sedimentary cores. The

aims of our study are (1) to capture the morphometric, cryolithologic and thermic properties of the Grøndalen pingo in comparison to other pingos on Spitsbergen and (2) to reconstruct the formation conditions of the pingo massive ice by applying stable isotope and hydrochemical approaches.



## 2 Study area

The Grøndalen study area on the western coast of Nordenskiöld Land (West Spitsbergen) is in about 10 km distance south-east of Barentsburg in whose vicinity the Russian Scientific Arctic Expedition on Spitsbergen Archipelago (RAE-S), maintains ground temperature and active-layer monitoring sites of the Global Terrestrial Network for Permafrost (GTN-P) and the Circumpolar Active Layer Monitoring (CALM) programs (Demidov et al., 2016; Christiansen et al., 2019). The Grøndalen is a trough valley in bedrock of Middle Jurassic–Palaeogene argillite and sandstone (Geological map Svalbard, 1991). Late Pleistocene and Holocene deposits in the lower part of Grøndalen reach more than 20 m thickness and likely represent sedimentation since at least the last glacial maximum (Verkulich et al., 2018). The mountains surrounding the valley reach up to 700-800 m above sea level (asl). The Grøn River is fed by many tributaries, which collect meltwater discharge from small hanging glaciers. In the upper valley part occur two larger glaciers, Tavlebreen and Passfjellbreen, as well as their terminal moraines.

The meteorological station in Barentsburg (WMO station #20107) at 75 m asl recorded a mean annual air temperature of −2.2 °C and a mean annual precipitation of 849 mm for 2016-2017. The ground temperatures reach –2.37 °C at 15 m depth below surface (bs) and the active layer thickness at the CALM site (3 km north of Barentsburg) measured in the end of September 2016 varied from 1.15 to 2.60 m with an average of 1.56 m (Demidov et al., 2016). Permafrost thickness in Barentsburg area varies along with morphology from 8-10 m near the seashore up to 300-450 m in the mountain and upland areas. According to measurements in the central part of Grøndalen the mean annual ground temperature amounts to -3.56 °C at depth of 15 m (borehole #8). Thus, the permafrost thickness probably exceeds 100 m below the valley surface.

In the central part of Grøndalen, a group of seven pingos occurs reaching diameters of 150 to 300 m and heights above their surroundings from 6.5 to 12.5 m (Fig. 1b). They are informally named as follows (west to east on Fig. 1b): Nori, Ori, Dori, Fili, Kili, Oin, Gloin after John R.R. Tolkien (1954-1955). The pingo Fili of 9.5 m maximum height (56 m asl) was chosen for drilling. It shows a clearly defined uppermost water-filled degradation crater of 5.5 m depth. The drilled Fili pingo is to the northeast and the southwest connected to Dori pingo and Kili pingo (Fig. 1b) and its slope shows radial dilation cracks.

## 3 Materials and methods

### 3.1 Drilling and ground temperature measurements

The drilling of the pingo performed in May 2017 reached a depth of 11.5 m bs (core #9, starting from 52.5 m asl, 77.99355 °N, 14.66211 °E), and was continued down to 25 m bs in April-May 2018. Additional cores were drilled on the pingo top reaching a depth of 12 m (core #10, from 56 m asl, 77.99332 °N, 14.66114 °E) and in the pingo surroundings reaching a depth of 6 m (core #11, from 47 m asl, 77.99531 °N, 14.66538 °E). Continuous temperature measurements in borehole #9 on Fili pingo were installed on 15 May 2018 using a 15-m long Geo Precision logger chain (M-Log5W cable) equipped with sensors every 0.75 m.





The cores were obtained with a portable gasoline-powered rotary drilling rig (UKB 12/25, Vorovskiy Machine Factory, Ekaterinburg, Russia) that allows performing operations without impact on the ecosystem. The device uses no drilling fluid, and relies on maintaining the frozen condition of the core for stratigraphic integrity and to prevent downhole contamination of the biogenic and sedimentological characteristics of the core. Drill diameters were 112 mm for the upper parts and 76 mm

for the lower ones. The core pieces were lifted to the surface every 30–50 cm. After documentation and cryolithological description (French and Shur, 2010) core pieces were sealed. Ice samples were kept frozen for transportation while sediment samples were kept unfrozen.

To obtain spring water samples at the foot of the Oin pingo near the right bank of the Grøn River drilling was performed through ca. 1 m of ice to pressurised water beneath an ice blister that formed at the spring source on 20 April 2018. Samples

were then taken of the water discharging through the 5 cm drill hole.

### 3.2 Mapping

The topographic mapping on 21 August 2018 covered an area of 0.25 km$^2$ using Global Navigation Satellite System (GNSS) Sokkia GRX2 devices and an Archer 2 base station. The obtained coordinates have planar accuracy of 1.5 cm and altitude accuracy of 2.5 cm.

### 3.3 Stable water isotopes

The concentration of water isotopes ($\delta D$ and $\delta^{18}O$) was measured at Climate and Environmental Research Laboratory (CERL, AARI St. Petersburg, Russia) using a Picarro L2120-$i$ analyzer. The working standard (SPB-2), measured after every five samples, was made of the distilled St. Petersburg tap water and calibrated against the IAEA standards VSMOW-2, GISP and SLAP-2. The reproducibility of results defined by re-measurements of randomly chosen samples was 0.08 ‰ for

$\delta^{18}O$ and 0.4 ‰ for $\delta D$, which is two orders of magnitude less than the common natural variability of the pingo ice isotopic composition and thus satisfactory for the purposes of the study. Additional samples from Grøndalen spring water were collected as unfiltered 20 ml aliquots in a screw-top HDPE bottle and analysed 6 times using a Picarro V 1102-I and a 2.5 µl injection volume with a precision error of 0.1 ‰ for $\delta^{18}O$ and 0.3 ‰ for $\delta D$. The $\delta^{18}O$ and $\delta D$ values are given as per mil (‰) difference to the Vienna Standard Mean Ocean Water (VSMOW) standard. The deuterium excess ($d$) is calculated as $d$

$= \delta D - 8\delta^{18}O$ (Dansgaard, 1964).

### 3.4 Hydrochemistry

The ion content of sedimentary permafrost samples of cores #9, #10 and #11 was estimated after water extraction at the analytical laboratory of RAE-S, Barentsburg. The material was dried and sieved at 1 mm. About 20 g of the sediment were suspended in 100 ml de-ionised water and filtered through 0.45 µm nylon mesh within 3 minutes after stirring. Electrical

conductivity (EC measured in µS cm$^{-1}$) and pH values were estimated with Mettler Toledo Seven Compact S 220. EC values were transformed automatically by the instrument in to general ion content values given as mg l$^{-1}$. Major anions and cations



in the water extracts were analysed by ion chromatography (Shimadzu LC-20 Prominence) equipped with the conductometric detector Shimadzu CDD-10AVvp and ion exchange columns for anions (Phenomenex Star-ion A300) and for cations (Shodex ICYS-50). Likewise melted pingo ice samples from core #9, snow and Grøn River water were analysed for pH, EC and ion composition after filtration through 0.45 µm nylon mesh. Spring water analysis of anions $Cl^-$, $NO_3^-$, $PO_4^{3-}$

and $SO_4^{2-}$ employed a Dionex ICS90 ion chromatography module calibrated in the range $0 - 2$ mg $l^{-1}$ for $NO_3^-$ and $PO_4^{3-}$ and in the range $0 - 50$ mg $l^{-1}$ for $Cl^-$ and $SO_4^{2-}$ (which required dilution). Precision errors were between 0.9 % ($SO_4^{2-}$) and 1.6 % ($PO_4^{3-}$), whilst the detection limit (three times the standard deviation of ten blanks) was $\leq 0.05$ mg $l^{-1}$. Alkalinity was deduced by headspace analysis of an acidified (pH 1.7) sample of 10 ml immediately after return to the laboratory using a PP Systems EGM 4 infra-red gas analyser (precision errors 4.5%). Immediately in the field, the spring water outflow was

analysed using Hach HQ40D meters for electrical conductivity and pH by gel electrode. To prevent the electrodes from freezing, this water was also injected by syringe into a pre-heated flow cell which maintained the water at about 7 °C. EC values in µS $cm^{-1}$ were transformed into general ion content values given as mg $l^{-1}$ by multiplication on 0.65.

## 4 Results

### 4.1 Ground temperature

The ground temperatures in borehole #9 on 12 September 2018 are shown in Fig. 2. At the lowermost sensor at 14.25 m bs the ground temperature reached –2.5 °C and varied at the same depth between –2.5 and –2.37 °C in the period from 15 May to 12 September 2018. The 0 °C point was observed at 1.5 m bs at the upper border of the massive pingo ice showing that the active-layer maximum depth reached the massive ice on 12 September 2018.

### 4.2 Cryolithology

**4.2.1 Core #9**

The 25 m long core #9 drilled from the pingo top crater exposed cover and basal sedimentary horizons enclosing massive pingo ice. From 0 to 1.5 m bs gravelly loam was found, which is assumed origin from the pingo top and moved downslope by cryoturbation and solifluction. Below this redeposited cover layer from 1.5 to 12 m bs transparent massive ice without any inclusions is observed. Air bubble content reaches up to 10 % in single ice layers. Between 12 and 22.2 m bs the pingo

ice remains transparent, but contains layers with 1-2 to 10-20 mm large dark silty flakes in subvertical orientation (up to 0.5 %). Alternating layers include rounded air bubbles (up to 10 %). The total thickness of the massive pingo ice amounts to 20.7 m. Its lower contact to the basal deposits is well defined in the core. From 22.2 to 25 m bs the massive pingo ice is underlain by dark grey clay with regular reticulate and irregular reticulate cryostructures (ice lenses 2 to 20 mm thick). At 22.3-23.5 m bs and at 23.7-23.8 m bs layers of clear (segregation) ice are found.

**4.2.2 Core #10**



The core #10 drilled on top of the pingo down to 12 m bs exposes sedimentary horizons. The uppermost part from 0 to 2.5 m bs includes the modern top soil at 0 to 0.1 m bs and a buried soil formation at 0.25 to 0.4 m bs both notable by higher contents of plant organic material. The minerogenic material is present by fine sand and loam including gravel. The cryostructures are wavy lenticular with ice lenses up to 2 cm thick. Toward 2.5 m bs the clay content increases as the gravel content decreases. From 2.5 to 12 m bs, the clay shows subhorizontal lenticular cryostructures up to 2 cm thick and includes in ice-oversaturated deposits and clear (segregation) ice at 4.7- 5.9 m bs, at 6.65-7.05 m bs and at 8.2-8.6 m bs although the massive ice of the pingo was not reached. At 5.9 to 6.2 m bs a layer of sand and gravel (up to 3 cm in diameter) with organic remains is observed.

### 4.2.3 Core #11

The core #11 was drilled in the surroundings of the pingo and reached a depth of 6 m bs, and is composed of gravelly sand and loam with structureless cryostructure.

### 4.3 Isotopic and hydrochemical properties of the massive pingo ice and spring water

According to trends in stable water isotopic composition of the massive ice obtained in core #9, four units are distinguished, which are unit I to unit IV (Fig. 2; Table 1).

Unit I (1.1-9.8 m bs) shows a down-core decreasing trend in $\delta^{18}O$ and $\delta D$ from $-9.6$ to $-16.8$ ‰ and from $-68$ to $-117$ ‰, respectively, while the $d$ increases from 8 to 18 ‰. Small inversions are notable at depths of 3.5 and 6.5 m bs (Fig. 2). The freezing of unit I is clearly expressed by $\delta^{18}O$-$\delta D$ and $\delta D$-$d$ slopes of 6.7 and $-0.2$, respectively (Fig. 3). The ion composition of unit I is dominated by $Na^+$ and $K^+$ in cations and $Cl^-$ and $HCO_3^-$ in anions. Variations in total ion content (mean 22 mg l$^{-1}$) are triggered by variations in $Na^+$ (0-24 mg l$^{-1}$), $Cl^-$ (1-14 mg l$^{-1}$) and $HCO_3^-$ (1-40 mg l$^{-1}$) concentrations (Fig. 2, Table 2) while $NO_3^-$ and $SO_4^{2-}$ contents vary little around 1 mg l$^{-1}$ each. $Ca^{2+}$ occurs solely in the lowermost part of unit I with low concentration. The pH varies between 6.6 and 7.9.

The isotopic composition of the pingo ice in unit II (9.8-16.1 m bs) exhibits down-core the opposite pattern as in unit I with increasing trend in $\delta^{18}O$ and $\delta D$ from $-16.7$ to $-11.1$ ‰ and from $-116$ to $-79$ ‰, respectively, while the $d$ decreases from 17 to 10 ‰ reaching almost the isotopic composition of the upper part of unit I (Fig. 2). The $\delta^{18}O$-$\delta D$ slope of 6.6 is slightly lower as in unit I while the $\delta D$-$d$ slope of $-0.2$ is very close to that of unit I (Fig. 3). The pH increases down-core from 7 to 8.8 as the ion content does from 14 to 138 mg l$^{-1}$ (mean 72 mg l$^{-1}$, Table 2). The latter correlates to increasing concentrations of $Na^+$, $Cl^-$, $SO_4^{2-}$ and $HCO_3^-$, while $NO_3^-$ remains almost stable and $K^+$ decreases (Fig. 2).

The stable isotope composition of unit III (16.1-20.8 m bs) resembles those of unit I with down-core decreasing $\delta^{18}O$ and $\delta D$ from $-10.8$ to $-15.2$ ‰ and from $-77$ to $-106$ ‰, respectively. The $d$ increases from 9 to 15 ‰. An inversion occurs at depth of 19.9 m bs (Fig. 2). The $\delta^{18}O$-$\delta D$ and $\delta D$-$d$ slopes are almost the same as in unit II with values of 6.6 and $-0.2$, respectively (Fig. 3). The ion content reaches highest values up to 428 mg l$^{-1}$ (mean of 202 mg l$^{-1}$, Table 2) in the lower part of unit III due to increased $Na^+$, $Cl^-$ and $HCO_3^-$ concentrations, where also Mg occurs (Fig. 2). The pH varies between 7.8 and 8.9.

The lowermost unit IV (20.8-22.2 m bs) shows down-core increase in $\delta^{18}O$ and $\delta D$ and decrease $d$ (Fig. 2) as unit II. The slopes of $\delta^{18}O$-$\delta D$ (–7.26) and $\delta D$-$d$ (–0.23) are the lowest all pingo ice units (Fig. 3). The mean ion content of 53 mg l$^{-1}$ is low (Table 2) due to largely reduced Na$^+$, K$^+$, HCO$_3^-$ and Cl$^-$ concentration (Fig. 2). The mean pH amounts to 7.8.

The overall pattern of the down-core isotopic composition of the pingo ice differentiates into to two modes of decreasing

$\delta^{18}O$ and $\delta D$ and increasing $d$ in units I and III and the opposite trends in units II and IV. The contents of Na$^+$, Cl$^-$ and HCO$_3^-$ ions increase from unit I to III, while K$^+$ decreases from unit I to II and increases again in unit III, NO$_3^-$ and SO$_4^{2-}$ show rather low variation, and Mg$^{2+}$ and Ca$^{2+}$ occur only occasionally with concentrations above 0.25 mg l$^{-1}$.

The spring water sampled in the vicinity of the Oin pingo in 2018 is characterized by $\delta^{18}O$ of –13.01 ‰, $\delta D$ of –93.5 ‰ and $d$ of 10.6 ‰ (Table 1). The ion content amounts to 1192 mg l$^{-1}$, with predominance of HCO$_3^-$ in anions (as indicated by the

high alkalinity 2.2 mM l$^{-1}$), while Cl$^-$ and SO$_4^{2-}$ are found in concentrations of 15.3 and 3.8 mg l$^{-1}$ respectively. The pH in the spring amounts to 8.8 (Table 2).

### 4.4 Hydrochemical properties of sedimentary water extracts

#### 4.4.1 Core #9

The hydrochemical signature of deposits underlying the massive ice of the pingo at 22.2 to 25 m bs is notable for high ion content reaching up to 1335 mg l$^{-1}$ and the prevalence of Na$^+$ and HCO$_3^-$ ions. (Fig. 4a, Table 3). Distinct ion content variations are driven by Na$^+$ and K$^+$ cations and HCO$_3^{2-}$ and Cl$^-$ anions. Ca$^{2+}$ and Mg$^{2+}$ have not been found and the anion composition resembles those of unit III. The pH is alkaline and varies between 9.2 and 9.9.

#### 4.4.2 Core #10

Core #10 drilled at the uppermost position of the pingo shows a different hydrochemical composition compared to the sedimentary water extracts than the bottom sediments of core #9 drilled nearby. The ion content is about nine times lower with mean values of about 99.6 mg l$^{-1}$ in core #10 and of about 1335 mg l$^{-1}$ in core #9 (Fig. 4a, b). The composition of major ions is also different with predominant SO$_4^{2-}$ and HCO$_3^-$ for anions while prevalent cations are still Na$^+$ and K$^+$. It is notable that Ca$^{2+}$ and Mg$^{2+}$ reach mean values of about 22 mg l$^{-1}$ and 2 mg l$^{-1}$, respectively (Table 3), pointing to the non-marine

origin of the deposits. The mean pH is neutral with 7.3±0.5.

#### 4.4.3 Core #11

The hydrochemical composition of major ions in core #11 generally resembles those of core #10 (Fig. 4b, c) although the ion content is much lower with a mean value of 38 mg l$^{-1}$ driven by more than two times lower SO$_4^{2-}$, ten times lower Cl$^-$, six times lower Na$^+$ and seven times lower Ca$^{2+}$ (Table 3). The pH is similar to those of core #10 with mean 7.2±0.3.



## 5 Discussion

### 5.1 Water sources of the massive pingo ice in Grøndalen

Following the current hypotheses on pingo growth on Spitsbergen, there are three possible main water sources for pingo massive ice formation, which are deep sub-permafrost groundwater for group I pingos connected to geologic fault structures,

sub-permafrost groundwater fed by sub-glacial melt for group II pingos and marine-originated groundwater for group III pingos (Matsuoka et al. 2004). Marine sources are assumed for hydraulic connections between groundwater of uplifted valleys after deglaciation and the sea (Yoshikawa and Harada, 1995). Sea ice, which is a result of seawater freezing is known to be sodium-chloride-dominated and shows ion contents between 2000 and 20000 mg l$^{-1}$ (Shokr and Sinha, 1995; Nazintsev and Panov, 2000) depending on freezing velocity, temperature and age. The massive ice of the pingo in Grøndalen shows

prevalent Na$^+$-HCO$_3^-$ contents and ion content with maximum values of 428 mg l$^{-1}$ (mean of 78±101 mg l$^{-1}$, Table 2). Thus, marine sources for the pingo massive ice are excluded and the previous assignment of the Grøndalen pingos to ancient group III pingos by Yoshikawa and Harada (1995) seems unlikely. Precipitation and surface waters in Grøndalen have lower ion contents and different composition if compared to the pingo massive ice (Table 2), which also excludes these sources as the main ones for the pingos of Grøndalen. The role of artesian flow resulting from the migration of sub-glacial groundwater for

pingo formation on Spitsbergen has been previously studied by Liestøl (1977, 1996) and Yoshikawa and Harada (1995). The original hydrochemical signature of water released from glacial melt is certainly altered by subsurface migration through various deposits. Pressurised spring water from beneath an ice blister sampled in 2018 shows predominance of HCO$_3^-$ in anions and ion content of 1192 mg l$^{-1}$ (Table 2). Furthermore, in 1921 the Norwegian geologist Anders K. Orvin described a spring in Grøndalen and sampled spring water in 1926 (Orvin, 1944), approximately at the northern side of today's Gloin

pingo (Fig. 1). This data from spring found in 2018 and 1926 seem meaningful to explore the potential water source for the pingo formations. The spring water analysed by Orvin was characterised by prevalent Na$^+$ and HCO$_3^-$ ions, which is typical for freshwater hydrogeological structures below permafrost (Romanovskiy, 1983), and an ion content of 879 mg l$^{-1}$ (Table 2). The sedimentary water-extract data from deposits underlying the massive ice shows likewise prevalent Na$^+$ and HCO$_3^-$ ions (Table 3). The similarity of the ion compositions of the spring water and the water extract of deposits underlying the massive

ice to those of the Fili pingo ice is striking although the absolute concentrations are much higher in the spring (1192 mg l$^{-1}$ in 2018 and 879.2 mg l$^{-1}$ in 1926) than in the massive ice (mean of all core#9 units 79.0±102.4). This difference between the ion concentrations in the spring water and in the pingo ice is most likely explained by salt expulsion during freezing, which is typical for ice formation from fresh ground water (Romanovskiy, 1983). We therefore conclude that the groundwater feeding the springs observed by Orvin (1944) and later within this study represents the main source water for the massive

pingo ice. This interpretation is further supported by the chain-like distribution of spring and pingos from east to west across the Grøndalen (Fig. 1b), which might delineate connectivity for sub-permafrost groundwater by the fault zone further protruding towards the Bøhmdalen.



It was suggested that groundwater recharge in Spitsbergen is related to the warm-based glaciers as there are no other potential taliks where groundwater recharge can occur (Orvin, 1944; Liestøl, 1976). Glaciers on the mounts of valley sides occur just several kilometres to the north from the Grøndalen pingos (Fig. 1), namely Irabreen and Stolleybreen. The water temperature is close to 0 °C when it is recharged below the glacier. Taking into account geothermal gradient of ~ 2 °C per 100 m in the coal survey wells in Barentsburg closest to the studied pingo (Ershov, 1998) and the fact that the base of surrounding glaciers is approximately at 500 m asl and pingos and spring are located between 100 and 50 m asl, geothermal heat will increase water temperature melted on the glacier base during its transit through the aquifer to the spring location directly to 9 °C as was observed by Orvin in 1921 (Orvin, 1944). Together with low salt content of the spring water, this observation allow suggesting that the source of sub-permafrost water discharged by the spring (and responsible for pingo formation) was sub-glacial melt heated and slightly salted during its rather fast transport through the aquifer to the discharge area.

## 5.2 Stable water isotope composition of the pingo massive ice in Grøndalen

Variations in the isotopic composition of the four massive pingo-ice units and between them might be explained by three main controls. Firstly, isotopic variations in the pingo ice might correspond to differing water sources migrating towards the freezing front during different periods of ice formation. Secondly, the source water of the pingo ice was constant, but had distinctly differing isotopic signatures over time. And thirdly, the isotopic variations of the pingo ice represent changes in closed, (semi-closed) and open system condition, i.e. freezing of a fixed or a (partly) renewed water volume. Estimations of pingo growth rate in Siberia and North America shows values of order decimetres per year (Chizhova and Vasil'chuk, 2018). That is why, concerning the first and the second controls, we assume rather no or only little changes in the water source and its isotopic composition over the rather small period of pingo formation. The isotopic composition of spring water sampled in Grøndalen in 2018 shows −13.01 ‰ in $\delta^{18}O$, −93.5 ‰ in $\delta D$ and a $d$ of 10.6 ‰, which is very close to the respective mean values of the massive pingo ice (all units) of −12.56±2.14 ‰ in $\delta^{18}O$, −89.0±14.2 ‰ in $\delta D$ and a $d$ of 11.5±4.1 ‰ (Table 1, Figure 3b, c). This observation makes it likely that the sub-permafrost groundwater feeding the spring also maintained the massive pingo-ice formation. Thus, the most probable explanation for the observed isotopic composition of the massive pingo ice relates to subsurface hydrologic conditions, i.e. the system state of closed or open conditions, which are further controlled by local groundwater pressure and the position of the permafrost table. Concurrent changes in the system character from closed to open as deduced from the isotopic and chemical compositions of the pingo ice units are outlined and discussed below.

The isotopic stratification of the massive pingo ice differentiates into four stages of pingo growth. The earliest stage of massive ice formation is represented in unit I (1.5-9.8 m bs). The down-core strong decline within unit I by about 7‰ $\delta^{18}O$, 49 ‰ in $\delta D$ accompanied by a strong rise of 10 ‰ in $d$ (Fig. 2) indicates a high near-surface temperature gradient, fast freezing and fast formation the about 8-m-thick unit I. If compared to a closed-system freezing model of Ekaykin et al. (2016), the down-core isotopic depletion in unit I is close to the modelled one (Fig. 5a) if an initial isotopic composition of −



12.6 ‰ in $\delta^{18}$O and −88 ‰ in $\delta$D is assumed. This is much lower than the modern mean annual precipitation values in the Barentsburg region with −9.0±4.2 ‰ in $\delta^{18}$O and −64±30 ‰ in $\delta$D, and closer to mean values of the Grøn River and its tributaries with about −11.7±0.3 ‰ in $\delta^{18}$O and −78±3 ‰ in $\delta$D (Table 1). Thus, if precipitation was the main source of the massive ice, its onset took place during a colder period than today. More likely, the water sources feeding the Grøn River

system, which are glacial meltwater and subsurface discharge represent the main water source of the massive pingo ice. According to the model, the lowermost (last formed) ice of unit I at a depth of 9.6 m bs corresponds to 85 % frozen water of the initial volume. If so the remaining unfrozen water would have had a highly depleted composition of about −19.0 ‰ in $\delta^{18}$O and −130 ‰ in $\delta$D (Fig. 5b). However, the modelled data do not entirely catch the real distribution of unit I isotopic composition where the $\delta^{18}$O values are slightly below and the $d$ values slightly above the respective modelled lines (Fig. 5b).

Because the model uses the maximum fractionation coefficient (Souchez and Jouzel, 1984), higher fractionation during freezing than modelled is impossible. Therefore, the system during freezing of unit I was likely not completely closed and new source water entered the system when about 50 % of the water was already frozen and changed the isotopic composition of the remaining unfrozen water. This is supported by the slight reversal in $\delta^{18}$O and $\delta$D, a more distinct reversal in $d$, and higher ion concentrations at depth of 6.5 m bs (Fig. 2). In Fig. 5a is shown that the last six most depleted $\delta^{18}$O data points

after about 50% of the water are frozen (corresponding to the lowermost data points of unit I in Fig. 2) increasingly deviate from the modelled data. Freezing of large parts of the unit I massive ice at least in only two stages under closed-system conditions is deduced.

The massive ice of unit II (9.8-16.1 m bs) is characterised by down-core increasing $\delta^{18}$O and $\delta$D by about 5.6 ‰ and 37 ‰, respectively. The $d$ decreases by about 8 ‰ (Fig. 2). Such down-core pattern might be explained by freezing under semi

closed-system conditions when the water reservoir got episodically renewed with isotopically less depleted subsurface water. The observed down-core increase in solute concentrations of the unit II ice (Fig. 2) further indicates ion enrichment in the source water during ongoing freezing, and therefore questions the occurrence of a completely open system fed by constant water supply.

The down-core isotopic composition of unit III (16.1-20.8 m bs) resembles those of unit I with declining $\delta^{18}$O by about 4.4

‰ and $\delta$D by about 29 ‰ concurrent with a rising $d$ by about 6 ‰ (Fig. 2). Reversals in $\delta^{18}$O and $\delta$D and corresponding $d$ are observed at depth of 19.9 m bs (Fig. 2), pointing to similar changes of the recharge reservoir as described for unit I. If compared to the freezing model under closed-system conditions (Fig. 5c) the water forming the ice of unit III would have had an initial composition of −14.4 ‰ in $\delta^{18}$O and −101 ‰ in $\delta$D. The more depleted isotopic values if compared to unit I are likely explained by fractionation of the available water volume due to previous freezing and formation of the older

massive ice units. The lowermost ice of unit II (−11.1 ‰ $\delta^{18}$O, −79 ‰ $\delta$D) is isotopically very close to the uppermost ice of unit III (− 10.8 ‰ $\delta^{18}$O, −77 ‰ $\delta$D), pointing to freezing of the same source water. The lowermost (last formed) ice of unit III at a depth of 20.5 m bs represents about 75 % frozen water of the initial reservoir (Fig. 5d) and points as the record of unit I again to freezing of a substantial part of the massive pingo ice (4.7 m thickness) of under prevailing closed-system



conditions. The down-core increase in solute concentrations with highest values in the lowermost part of unit III supports closed-system freezing conditions of a fixed water volume.

The lowermost unit IV (20.8-22.2 m bs) exhibits down-core increasing $\delta^{18}O$ by about 2 ‰ and $\delta D$ by about 15 ‰ while $d$ decreases by about 3 ‰ similarly to unit II but at less ranges. A drop in solute concentrations is striking (Fig. 2). Both, hydrochemical and isotopic composition of unit IV point to freezing conditions of a semi-closed system.

In summary, the pingo ice record obtained in core #9 delineates two closed-system freezing episodes (units I and III) with only slight recharge inversions of the water reservoir and two episodes (units II and IV) with more complicated freezing of subsurface water under semi-closed conditions when the reservoir got renewed from the same source. A changing water source from less glacial runoff- to more precipitation-driven water sources might also induce the observed down-core increase in $\delta^{18}O$ and $\delta D$ and the accompanying decrease in $d$. Generally increasing $\delta^{18}O$ and $\delta D$ values are not associated with changes in the freezing that produces constant to decreasing values from open to closed system.

Nowadays, the pingo has already accomplished its active growth as seen by the degradation crater on top of the pingo and the occurrence frozen deposits underlying the massive pingo ice. The latter induces that the freezing front where the pingo ice formed disappeared when the ground got deeply frozen. The relatively warm ground temperature of only –2.5 °C at 14.25 m bs in borehole #9 and the active layer reaching the top of the massive ice lead to its successive melt and intensified solifluction further lowering the thickness of the protective layer above the massive ice. As a further consequence of ongoing degradation of the pingo, the crater lake (20 x 30 m) might develop into a larger thermokarst lake as the massive ice melt proceeds. Such solifluction and thermokarst degradation processes are common for Spitsbergen pingos (Liestøl, 1996).

**5.3 Some aspects of pingo formation in the context of Grøndalen valley history**

The elongated outer shape of the Fili pingo cone is also observed for other pingos on Spitsbergen (Liestøl, 1996) while the revealed information on the internal structure of Fili pingo is unique in the sense that it is the only pingo completely drilled in its centre. The observations of the Riverbed pingo in Adventdalen where river erosion removed the slope deposits and exposed ice, gravel and sand in a distal position of the pingo (Matsuoka et al., 2004) are, however, not directly comparable in exposure and  internal structure to those of Fili pingo that exhibits the entire massive ice in its central position.

The shape of the massive pingo ice can be deduced in vertical extension from its upper and lower boundaries observed in core #9 between 1.5 and 22.2 m bs. In lateral extension, the core #10 drilled from the crater top down to 12 m bs in about 35 m distance from the core #9 position did not reach the massive ice and suggests a rather steep slope of the massive ice body. The assumed shape of the pingo and its massive ice is shown in Fig. 1c. The thickness of the massive ice (20.7 m) exceeds the height of the pingo of 9.5 m, which is explained by step-wise massive ice growth when the ongoing subsurface freezing pushed the previously formed ice and cover deposits upward. The latter moved subsequently from the pingo top down-slope by solifluction. Thus, the 9.5 m amplitude of surface uplift seen in the modern stage of pingo evolution became less than the 20.7 m thickness of the massive ice. Solifluction further explains the presence of the buried soil observed at 0.25-0.4 m bs in core #10. If these assumptions are reliable, the massive ice formation started at the freezing front at a depth of about 15 m bs.



Because the ongoing freezing of new ice below the previously formed massive ice is only possible at the contact between ice and unfrozen waterlogged deposits, it seems likely that the advance of the freezing front got compensated by the growth of the massive ice and the pingo heave as well as by geothermal heat transported by groundwater. Consequently, the ground temperature at the base of the massive ice remained at the freezing point during the pingo growth. Even if the advance of the

freezing front slowed somewhat down by massive ice formation, pingo growth and geothermal heat from groundwater, the latter did not slow down the freezing of the cover deposits along the shape of the aggrading hill, which likely promoted lateral freezing from the slopes, and terminated pingo growth.

Because information on the origin and ages of the deposits surrounding the pingos of Grøndalen are still lacking, such aspects will be addressed in future work. Nevertheless, tentative assumptions of the pingo formation stages in context of the

valley evolution can already been drawn. While pingo formation in general depends on freezing of loose deposits and water migration towards the freezing front, the presence of pingos in Grøndalen confirm epigenetic freezing of formerly unfrozen deposits. Yoshikawa and Harada (1995) conclude from the position of the Grøndalen pingos at about 50 m asl that their formation started quickly after retreat of the sea (Fig. 6).

After retreat of the sea and establishment of the Grøn River system, we assume a sedimentation period of non-marine

gravelly sand and loam deposits observed in core #11. These deposits also covers the top and the slopes of the Fili pingo (core #10) and represent therefore the non-marine ground in which the Grøndalen pingos formed, contradicting the interpretation of Yoshikawa and Harada (1995) who proposed pingo growth within refreezing marine sediments after sea regression. Based on the finding of lower ice body boundary at the depth 15 m under the surrounding surface (Fig. 1c), the position of the freezing front is assumed to have reached this depth before start of pingo growth. To reach such depth, we

further assume a certain period of time that was also needed to disconnect the groundwater hydrology of the valley from the sea since seawater is unlikely to have been the source of the pingo massive ice as discussed above. The aggrading permafrost in Grøndalen likely restructured the groundwater hydrogeology of the valley and created groundwater flow in the fault zone connected to Bøhmdalen that fed the Grøndalen pingo and explains their chain-like occurrence (Fig. 1b). Comparable to the pingo formation in Adventdalen, where the oldest pingos developed at higher positions (Fig. 6b), more distant from the sea

(Yoshikawa and Nakamura, 1996), we propose a similar pattern in Grøndalen. If this assumption is correct, Nori pingo is the youngest and Gloin pingo is the oldest in the pingo chain of Grøndalen (Fig. 1b). While the taliks feeding the oldest pingos (above 50 m asl) froze subsequently, the next connectivity at lower topography became the place for new pingo formation and thus creating a chain of pingos following fault zone and the downslope topographic gradient and hence the availability of groundwater (Fig. 6c). The activity of groundwater springs related to warm-based glaciers might reflect their shrinking by

less discharge of even disappearance of the springs (Haldersen et al., 2011). This is because glacier size decrease and surface lowering induce shrinking accumulation area and further decreasing warm-based area terminating the recharge of springs. In this context, in the study of Chernov and Muraviev (2018) it was shown that the loss in glacier area in Nordenskiöld Land (West Spitsbergen) between 1936 and 2017 reached 49.5 %. Water discharge by the spring in Grøndalen does not necessarily means that surrounding glaciers are warm-based at current time. They are probably cold-based today because of





their current small size, but meltwater from the stage when they were warm-based may still be discharged by Grøndalen springs due to low discharge rates at rare locations. Pingos in Grøndalen accomplished their growth not because of stop of ground water flow but because of freezing of sediments below the massive ice inhabiting the water supply. Today Grøndalen pingos exhibit clear degradation features such as the crater lake on top of Fili pingo (Fig. 6d).

## 6 Conclusions

For the first time a pingo on Spitsbergen was completely drilled to obtain records of the massive ice and the deposits above and the permafrost below it. The massive pingo ice is almost clear and reaches a thickness of 20.7 m while the pingo reaches a height above the surrounding surface of 9.5 m at present. Both, pingo ice thickness and pingo height were reduced by degradation. The lowermost measured ground temperature at 14.25 m bs close to the zero-amplitude temperature showed only little variation between –2.5 and –2.37 °C from May to September 2018. The active layer depth of 1.5 m in September 2018 reached the uppermost massive ice, which indicates the ongoing fast degradation of the pingo. This is further seen in the crater lake on top of the pingo and strong solifluction that removes cover deposits downslope.

The stable water isotope record of the massive ice shows two episodes of closed-system freezing and two episodes of semi-closed freezing when the source water feeding the massive ice formation recharged. The hydrochemical composition of the massive ice and the permafrost below is dominated by $Na^+$ and $HCO_3^-$ ions, thus of terrestrial origin and similar to those of historical and modern observations of spring water in Grøndalen.

Our current understanding of pingo-related processes and conditions in Grøndalen makes it difficult to align them to the pingo categories for Spitsbergen proposed by Yoshikawa and Harada (1995). We identified characteristics such as fault-related groundwater discharge (aligned to group I) and ground-water origin from warm-based glaciers (aligned to group II) while their formation within epigenetically refreezing marine deposits immediately after sea regression (aligned to group III) seem unlikely due to the non-marine character of the deposits surrounding the Grøndalen pingos.

Pingos are highly sensitive to climate warming, there origin and distribution over Spitsbergen valleys depends on complex interaction of hydrogeological structures with climate, sea, glaciers, permafrost and, thus their further investigation is one of the keys to understanding history of climate evolution.

## Author contributions

ND and SV initiated and designed the present study. They further drilled and documented the cores together with VD. AE carried out stable isotope analyses. AE and AJH contributed hydrochemical and stable isotope data of the modern environment, such as from precipitation, surface waters and sources. MA, LS and HM supported the overall data analysis and interpretation. ND and SW wrote the paper with input from the other co-authors, who contributed equally to the final discussion of the results and interpretations.



## Competing interests

The authors declare that they have no conflict of interest.

## Acknowledgements

We acknowledge support for field logistics and lab analytics from the Russian Scientific Arctic Expedition on Spitsbergen Archipelago (RAE-S), Barentsburg. Ekaterina Poliakova (Arctic and Antarctic Research Institute, St. Petersburg, Russia) provided useful translations of Norwegian original literature. SW was supported by Deutsche Forschungsgemeinschaft (grant no. WE4390/7-1). AH acknowledges JPI-Climate Topic 2: Russian Arctic and Boreal Systems, Award No. 71126
(LowPerm).

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



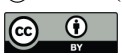

**Tables**

Table 1: Stable isotope (δ¹⁸O, δD, and *d*) minimum, mean, maximum values, standard deviations (std), and slopes, intercept and correlation coefficient (r²) of δ¹⁸O-δD and δD-*d* plots from massive pingo ice of core #9, from intrasedimental ice of core #10 and from precipitation (2016-2018). Data are shown in Fig. 3. Additional data from 2018 include spring water of Grøndalen and river and tributary water of the Grøn River (Skakun et al., in review).

| Material | | core #9 unit I | core #9 unit II | core #9 unit III | core #9 unit IV | core #9 units I-IV | core #10 | Modern precipitation | Grøn River and tributaries | Grøndalen spring |
|---|---|---|---|---|---|---|---|---|---|---|
| Core depth | [m bs] | 1.5-9.8 | 9.8-16.1 | 16.1-20.8 | 20.8-22.2 | 1.4-22.2 | 6.65-8.4 | surface | surface | surface |
| n | | 16 | 10 | 7 | 3 | 38 | 4 | 88 | 7 | 1 |
| δ¹⁸O min | [‰] | −16.81 | −16.65 | −15.18 | −12.80 | −16.81 | −18.93 | −18.42 | −11.71 | -- |
| δ¹⁸O mean | [‰] | −12.56 | −13.06 | −13.17 | −11.42 | −12.56 | −14.09 | −8.95 | −11.24 | −13.01 |
| δ¹⁸O max | [‰] | −9.56 | −11.07 | −10.80 | −10.60 | −9.52 | −11.98 | −0.80 | −10.89 | -- |
| δ¹⁸O std | [‰] | 2.26 | 2.34 | 1.58 | 1.20 | 2.14 | 3.25 | 4.23 | 0.27 | -- |
| δD min | [‰] | −116.9 | −115.9 | −106.3 | −90.7 | −116.9 | −131.3 | −144.5 | −83.2 | -- |
| δD mean | [‰] | −88.9 | −92.0 | −93.1 | −81.7 | −89.0 | −98.8 | −64.2 | −78.1 | −93.5 |
| δD max | [‰] | −68.4 | −78.8 | −77.2 | −6.1 | −68.4 | −84.4 | −17.3 | −75.2 | -- |
| δD std | [‰] | 15.1 | 15.4 | 10.5 | 7.9 | 14.2 | 21.9 | 30.2 | 2.6 | -- |
| *d* min | [‰] | 8.0 | 9.7 | 9.2 | 8.5 | 7.6 | 11.5 | −25.0 | 10.5 | -- |
| *d* mean | [‰] | 11.5 | 12.4 | 12.3 | 9.7 | 11.5 | 13.9 | 7.5 | 11.9 | 10.6 |
| *d* max | [‰] | 17.6 | 17.4 | 15.1 | 11.7 | 17.6 | 20.1 | 45.1 | 13.3 | -- |
| *d* std | [‰] | 3.1 | 3.3 | 2.2 | 1.8 | 2.9 | 4.1 | 10.8 | 1.3 | -- |
| Slope | δ¹⁸O-δD | 6.66 | 6.60 | 6.58 | 6.52 | 6.63 | 6.74 | 6.78 | 8.18 | -- |
| Intercept | δ¹⁸O-δD | −5.28 | −5.88 | −6.46 | −7.26 | −5.67 | −3.81 | −3.39 | +13.92 | -- |
| r² | δ¹⁸O-δD | 1 | 1 | 1 | 1 | 1 | 1 | 0.90 | 0.75 | -- |
| Slope | δD-*d* | −0.20 | −0.21 | −0.21 | −0.23 | −0.21 | -- | -- | -- | -- |
| Intercept | δD-*d* | −6.26 | −7.10 | −7.72 | −8.78 | −6.77 | -- | -- | -- | -- |
| r² | δD-*d* | 0.97 | 0.99 | 0.96 | 0.97 | 0.98 | -- | -- | -- | -- |



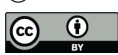

Table 2: Hydrochemical composition of the massive pingo ice of core #9 given as minimum, mean, maximum values, standard deviations (std) per unit. Data are shown in Fig. 2. Additional data include spring water of Grøndalen from 1926 (Orvin, 1944) and 2018 (this study), snow from 2016-2018 and river and tributary water of the Grøn River (Skakun et al., in review).

| Material | | core#9 unit I | core#9 unit II | core#9 unit III | core#9 unit IV | core#9 all units | Grøn River | Snow | Spring 1926 | Spring 2018 |
|---|---|---|---|---|---|---|---|---|---|---|
| Core depth | [m bs] | 1.5-9.8 | 9.8-16.1 | 16.1-20.8 | 20.8-22.2 | 1.5-22.2 | surface | surface | surface | surface |
| n | | 16 | 7 | 8 | 3 | 34 | 1 | 1 | 1 | 1 |
| pH min-max | | 6.6-7.9 | 7.0-8.8 | 7.8-8.9 | 7.7-7.9 | 6.6-8.9 | -- | -- | -- | -- |
| pH mean | | 7.2±0.4 | 7.8±0.7 | 8.4±0.4 | 7.8±0.1 | 7.7±0.7 | 7.4 | 7.7 | -- | 8.8 |
| EC min-max | [µS cm⁻¹] | 5-106 | 27-269 | 133-834 | 100-108 | 5-834 | -- | -- | -- | -- |
| EC mean | [µS cm⁻¹] | 44±30 | 141±118 | 394±265 | 104±5 | 152±197 | -- | -- | -- | 1834 |
| ion content min-max | [mg l⁻¹] | 2.6-54.2 | 13.8-137.9 | 68.1-427.7 | 51.0-55.6 | 2.6-427.7 | -- | -- | -- | -- |
| ion content mean | [mg l⁻¹] | 22.3±14.9 | 72.2±60.5 | 202.0±136.0 | 53.3±2.3 | 77.8±100.9 | 118.4 | 16.4 | 879.2 | 1192 |
| Na⁺ min-max | [mg l⁻¹] | 0.0-23.8 | 5.4-70.3 | 39.8-216.9 | 25.6-28.4 | 0.0-216.9 | -- | -- | -- | -- |
| Na⁺ mean | [mg l⁻¹] | 8.2±6.8 | 30.8±25.2 | 107.3±68.1 | 27.2±1.5 | 37.8±52.3 | 7.6 | 5.3 | 333.3 | -- |
| K⁺ min-max | [mg l⁻¹] | 1.7-4.1 | 0.3-2.1 | 0.2-2.8 | 0.2-0.3 | 0.2-4.1 | -- | -- | -- | -- |
| K⁺ mean | [mg l⁻¹] | 2.5±0.7 | 1.1±0.7 | 1.1±1.0 | 0.3±0.1 | 1.7±1.1 | 0.9 | < 0.25 | 16.5 | -- |
| Ca²⁺ min-max | [mg l⁻¹] | 0.0-0.5 | 0.0-0.2 | -- | -- | 0.0-0.5 | -- | -- | -- | -- |
| Ca²⁺ mean | [mg l⁻¹] | -- | -- | -- | -- | -- | 13.0 | 0.3 | 4.9 | -- |
| Mg²⁺ min-max | [mg l⁻¹] | -- | -- | 0.0-1.9 | -- | 0.0-1.9 | -- | -- | -- | -- |
| Mg²⁺ mean | [mg l⁻¹] | -- | -- | -- | -- | -- | 9.1 | 0.3 | 6.1 | -- |
| Cl⁻ min-max | [mg l⁻¹] | 1.1-14.4 | 4.3-25.1 | 14.3-120.7 | 8.8-12.4 | 1.1-120.7 | -- | -- | -- | -- |
| Cl⁻ mean | [mg l⁻¹] | 6.4±4.1 | 12.2±7.8 | 51.1±40.3 | 11.0±1.9 | 18.5±26.6 | 5.3 | 8.0 | 113.5 | 15.3 |
| SO₄²⁻ min-max | [mg l⁻¹] | 0.6-1.8 | 0.7-4.4 | 1.4-3.1 | 1.3-1.8 | 0.6-4.4 | -- | -- | -- | -- |
| SO₄²⁻ mean | [mg l⁻¹] | 0.9±0.3 | 1.8±1.3 | 2.1±0.6 | 1.5±0.3 | 1.4±0.9 | 63.3 | 0.6 | 3.3 | 3.8 |
| HCO₃²⁻ min-max | [mg l⁻¹] | 1.0-39.9 | 6.9-110.1 | 56.4-336.3 | 39.5-42.5 | 1.0-336.3 | -- | -- | -- | -- |
| HCO₃²⁻ mean | [mg l⁻¹] | 13.5±11.5 | 51.6±41.8 | 163.3±106.8 | 40.5±1.7 | 59.2±81.4 | 19.2 | 2.0 | 372.3 | -- |
| NO₃⁻ min-max | [mg l⁻¹] | 0.7-1.1 | 0.5-0.7 | 0.5-2.1 | 0.6-0.7 | 0.5-2.1 | -- | -- | -- | -- |
| NO₃⁻ mean | [mg l⁻¹] | 0.7±0.1 | 0.6±0.1 | 0.8±0.5 | 0.7±0.1 | 0.7±0.3 | -- | -- | -- | -- |
| Alkalinity | [mM] | -- | -- | -- | -- | -- | -- | -- | -- | 2.19 |



**Table 3: Hydrochemical composition of water extracts of the sedimentary cores #9, #10 and #11 given as minimum, mean, maximum values, standard deviations (std) per core. Data are shown in Fig. 4.**

| Sedimentary water extract | | core #9 | core #10 | core #11 |
|---|---|---|---|---|
| Core depth | [m bs] | 22.55-24.9 | 0-11.9 | 0-5.0 |
| n | | 5 | 10 | 4 |
| pH min-max | | 9.2-9.9 | 6.7-7.9 | 6.9-7.6 |
| pH mean | | 9.6±0.2 | 7.3±0.5 | 7.2±0.3 |
| EC min-max | [µS cm⁻¹] | 1011-2670 | 22-436 | 14-176 |
| EC mean | [µS cm⁻¹] | 1796±818 | 199±120 | 75±71 |
| ion content min-max | [mg l⁻¹] | 505.0-1335.0 | 10.8-218.0 | 6.9-88.1 |
| ion content mean | [mg l⁻¹] | 897.8±408.8 | 99.6±60.0 | 37.5±35.6 |
| Na⁺ min-max | [mg l⁻¹] | 429.2-861.6 | 3.9-81.1 | 2.5-10.3 |
| Na⁺ mean | [mg l⁻¹] | 625.9±204.4 | 46.4±28.1 | 8.0±3.7 |
| K⁺ min-max | [mg l⁻¹] | 14.4-22.6 | 1.3-15.4 | 0.6-5.1 |
| K⁺ mean | [mg l⁻¹] | 18.5±3.7 | 8.0±4.2 | 2.6±2.3 |
| Ca²⁺ min-max | [mg l⁻¹] | -- | 0.6-49.0 | 1.0-15.7 |
| Ca²⁺ mean | [mg l⁻¹] | -- | 22.2±17.0 | 6.2±8.2 |
| Mg²⁺ min-max | [mg l⁻¹] | -- | 0.5-4.1 | 0.3-6.4 |
| Mg²⁺ mean | [mg l⁻¹] | -- | 2.0±1.3 | 2.6±3.3 |
| Cl⁻ min-max | [mg l⁻¹] | 107.0-397.2 | 0.0-31.6 | 0.8-2.4 |
| Cl⁻ mean | [mg l⁻¹] | 216.0±144.7 | 15.6±10.4 | 1.6±0.7 |
| SO₄²⁻ min-max | [mg l⁻¹] | 52.6-93.8 | 1.4-138.0 | 1.4-50.1 |
| SO₄²⁻ mean | [mg l⁻¹] | 74.7±14.9 | 47.4±38.9 | 18.3±22.4 |
| HCO₃²⁻ min-max | [mg l⁻¹] | 289.0-836.8 | 4.2-49.0 | 4.2-40.3 |
| HCO₃²⁻ mean | [mg l⁻¹] | 522.9±227.5 | 22.2±17.0 | 15.7±16.6 |
| NO₃⁻ min-max | [mg l⁻¹] | 0.6-0.9 | 0.1-0.4 | 0.1-0.3 |
| NO₃⁻ mean | [mg l⁻¹] | 0.7±0.1 | 0.2±0.1 | 0.2±0.1 |





**Figures**



Figure 1: Study region of Nordenskiöld Land on West Spitsbergen (inset), showing (a) the position of Grøndalen, (b) the position of seven pingos in Grøndalen (redrawn after https://toposvalbard.npolar.no) and (c) the drilling profile across Fili pingo (shown as orange star in b) with locations of cores #9, #10 and #11.




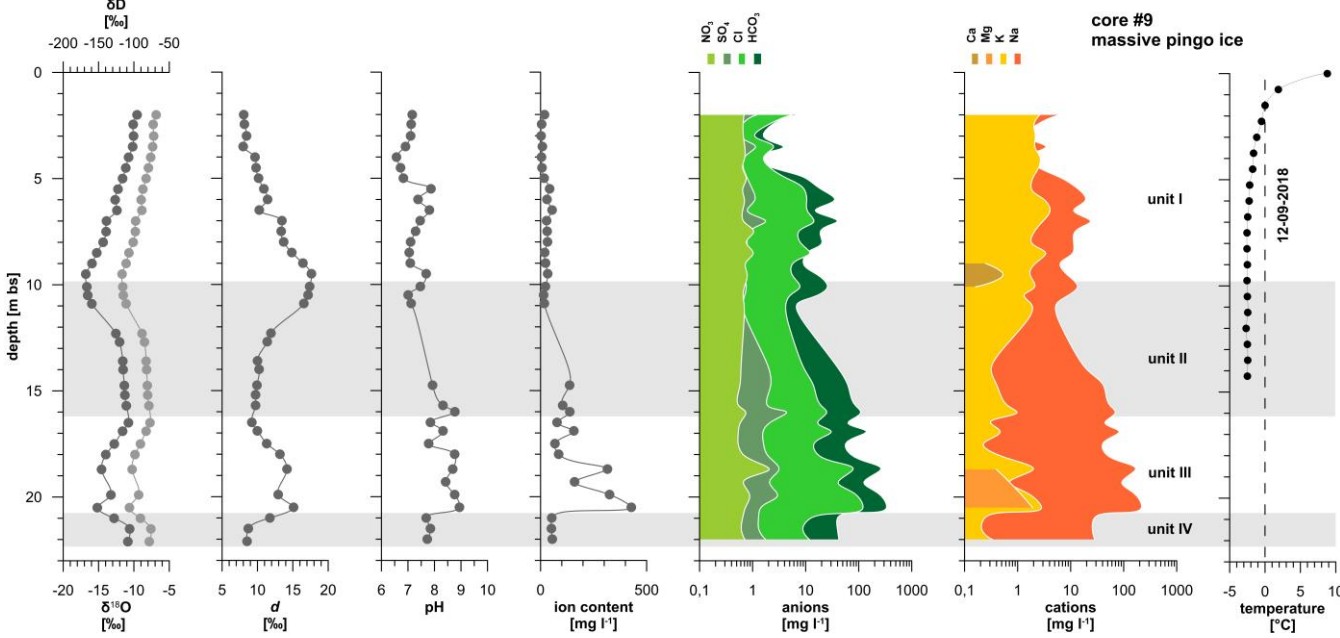

**Figure 2: Isotopic and hydrochemical composition of the massive ice of Fili pingo obtained from core#9 as well as thermometric data from the borehole on 12 September 2018. Light grey symbols in the first plot refer to the upper x-axis (δD). Data are given in Table 1 and Table 2.**



The Cryosphere



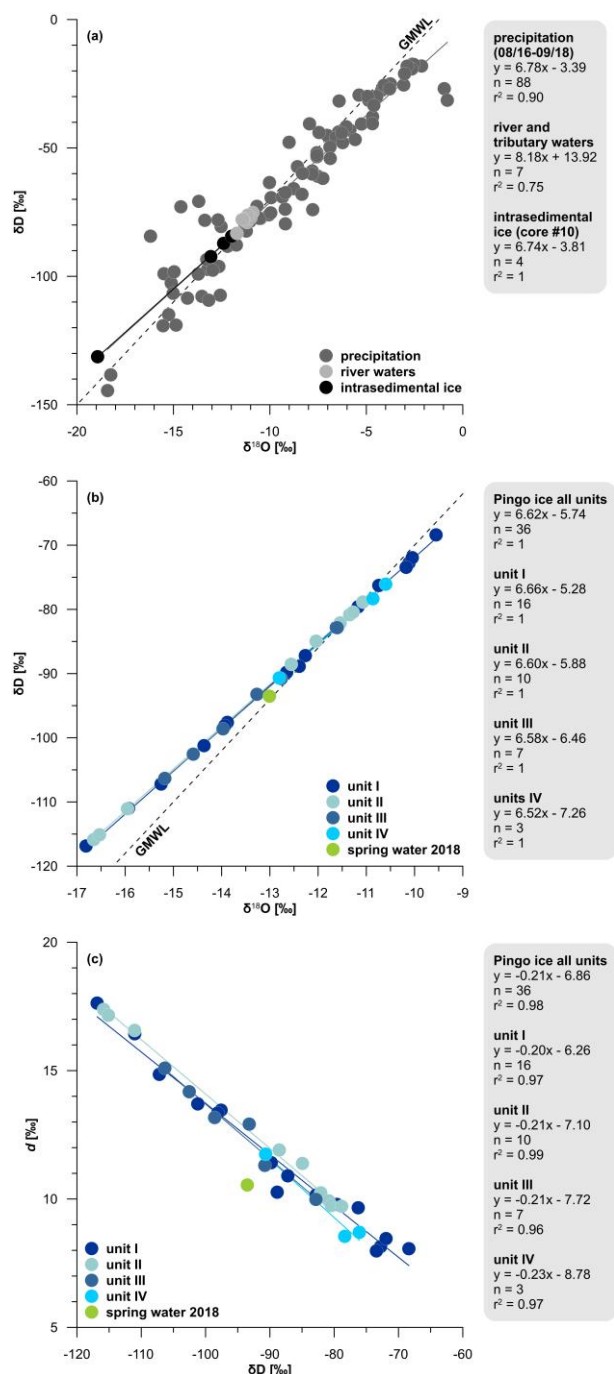

**Figure 3: Co-isotopic plots of (a) δ$^{18}$O and δD in modern precipitation (08/2016-09/2018) and water of the Grøn River and its tributaries (Shakun et al., in review), (b) δ$^{18}$O and δD in massive ice of core #9 from pingo and from spring water sampled, and (c) δD and deuterium excess (*d*) data in massive ice of core #9 from pingo and from spring water. Data are given in Table 1. Note different axis scales in (a) and (b).**



**Figure 4: Hydrochemical composition of water extracts from the sedimentary cores #9, #10 and #11. Data are given in Table 3.**
**Note different axis scale for ion content in (a) reaching up to 1335 mg l⁻¹.**



**Figure 5: Isotopic composition of unit I (a, b) and unit III (c, d) of the massive pingo ice of core #9 in comparison to freezing model data under closed-system conditions (Ekaykin et al., 2016).**



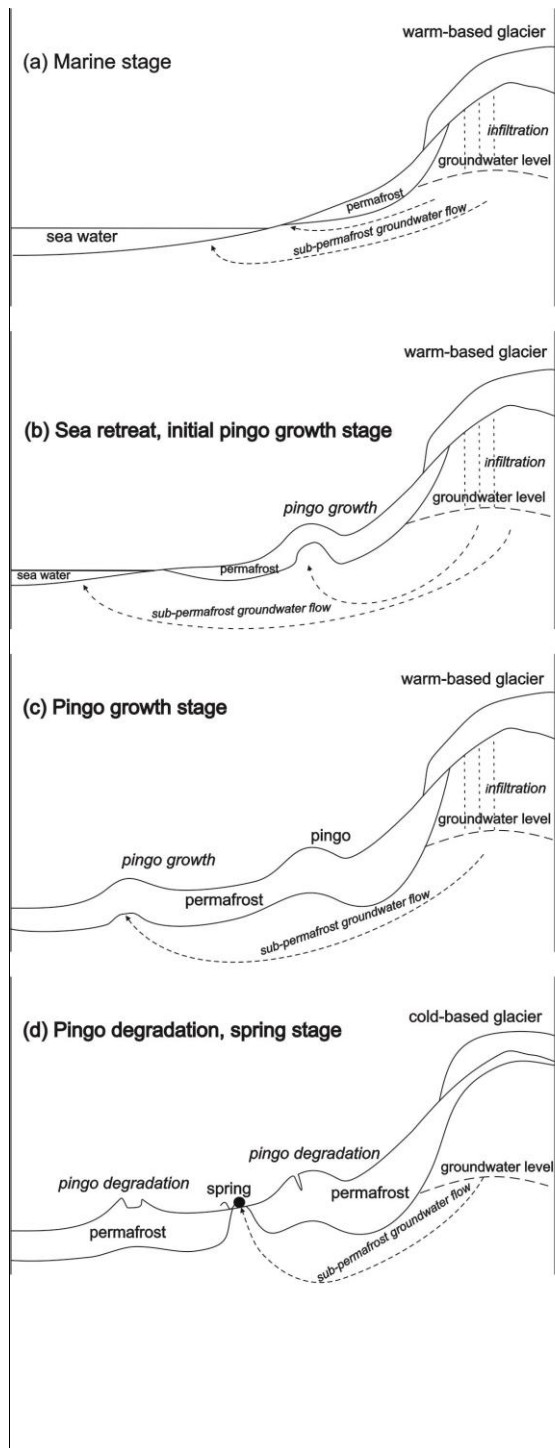

**Figure 6: Schematic sketch of Grøndalen evolution and pingo formation differentiating into (a) marine stage, (b) initial pingo growth after sea retreat and establishment of the valley's hydrological system, (c) continuing pingo growth along the topographic gradient and (d) current pingo degradation and occurrence of springs.**