# Peer review of "Geochemical signatures of pingo ice and its origin in Grøndalen, West Spitsbergen"

_The Cryosphere, 2019_

## Referee Comment (RC1) · Trevor Porter (Referee) · 29 May 2019

The authors present stratigraphic profiles of solutes and water isotopes from pore ice of permafrost cores collected from the Fili pingo in West Spitsbergen, and of modern precipitation, tributaries, and a local spring to better constrain the origin of water in the pingo system. Based on the data, the authors deduce that the pingo is spring-fed, and that its evolution was characterised by several distinct periods of closed and semi-closed conditions, as evidenced by trends in the water isotope and solute data. They also use a Rayleigh isotope distillation model to show that the data diverge from a closed-system. I found the methods and interpretation were robust. The paper was well written, easy to follow and the topic is well suited for The Cryosphere. Below are several comments meant to help the authors improve the communication of their work,

and address one uncertainty that is not discussed. Following these minor revisions I would recommend this paper for publication.

Major comments. I am intrigued by the vertical trends in the water isotope dataset. To a large extent I agree with the interpretation assuming the following conditions: core #9 was drilled exactly in the centroid of the ice body; pingo geometry is conical; and it is reasonable to assume the pingo grew equally on all sides. Largely these points are not discussed. Isotopic stratigraphy of pingo ice could show 'apparent' reversals if the coring angle was off-axis, or pingo growth was asymmetric. Perhaps the authors can comment on this uncertainty. Again, I am in agreement with the interpretation, but would appreciate if this issue of coring angle and pingo growth geometry could be discussed.

Minor comments. Abstract. The final 2 sentences are largely unconnected to the research. Please finish the abstract with some kind of significance statement instead.

P2, L20-22. this sentence is too wordy, and confuses the message. please make it more concise.

P2, L24-26. please elaborate on why this is true.

P3, L26-27. the description of core thickness in relation to different baselevel elevations is a bit confusing. Please clarify in simple terms.

P6, L17. the reported dD-d18O and d-dD slopes (6.7 and -0.2, respectively) are nearly identical to the slopes observed in modern precipitation. Fig. 3 indicates dD-d18O slope is 6.78, and based on the dD-d18O equation the d-dD slope can be calculated to -0.18 (or -0.2 if rounded to 1 decimal). My point is, your claim that the effects of freezing on the co-isotope slopes are not well supported by the data, since precipitation has these slopes.

P9, L19-20. it is unclear how the previous sentence justifies this conclusion. please elaborate.

P10, L12-13. If true, you may be able to calculate the rate of pore water recharge based on deviation from isotope distillation model.

P12, L10. ...valley evolution can already 'be' drawn.

Figures. The font size and resolution of some of the figures is too low for publication, and in some cases it was difficult to interpret the figures as given. Please revise to conform to the publication standards of The Cryosphere.

---

## Referee Comment (RC2) · Go Iwahana (Referee) · 27 Jun 2019

The paper "Pingo development in Grondalen, West Spitsbergen" by Demidov et al. presents very rare data set about internal structure and ice geochemistry of a pingo in West Spitsbergern. The paper discusses possible water sources and freezing conditions of the core ice, then growth history the pingos in target area. Especially, the cryolithological information with geochemistry of entire pingo core ice (with underlain sediment) is a paramount value for understanding frozen ground on Earth and other planets. This paper should be published ultimately with additional information after some clarifications of information provided and revision of discussion. (Major comments) The title of paper is too broad, and it indicates overall study on pingo distribution and development history in Grondalen. However, the focus of this paper, to me, is

unveiled internal structure of this particular pingo and interpretation of geochemistry of pingo ice and surrounding water. Could you revise the title so that reader can easily understand the contents of the paper more specifically?

Discussion about water source of pingo ice is not clear mainly because the definition of precipitation is vague in the text. I encourage authors to reconstruct the discussion considering time and area of the precipitation and groundwater. I think the confusion came from the fact that the water sources of pingo ice and sources of groundwater (and river) are different concepts (groundwater can be a source of pingo ice, but the groundwater itself has its water sources.).

Section 5.3 should be rewritten and revise thoroughly to clearly present authors' discussion. Its paragraph structure does not match discussion flows. It was very hard to follow the logic of authors' idea and some statements don't sound to me as pointed in minor comments below. Authors discussed hydrologic conditions and history of pingo growth comparing to Yoshikawa and Harada (1995) model; however, explanation, evidence and reasoning to suggestion of non-marine sedimentation are weak and discrepancy points between researches are unclear. This can be improved by describing more details about Yoshikawa and Harada (1995)'s development model and their reasoning if you intend to include this comparison in the conclusion. Please make it clear about discrepancies and discussion in occurrence of sea regression at the target pingo location, timing of the regression, interpretation of sedimentation history at the site, and judgment of marine or non-marine sediment.

I think it is important to show photo of obtained cores for judgment of integrity, and also to capture cryolithologic properties of the target pingo as authors indicated as one of the purposes. Recovery of the entire massive ice core of a pingo is a dominant value of this study, however, cryolithological description and discussion of the obtained core is poor. Aim (1) can be more developed by comparing to other pingos on Spitsbergen (and in other regions?).

(Specific minor comments) P2, first paragraph: please provide some references for these descriptions. P2, L25: What are the unresolved questions relevant to this paper? P3, L23: I assume 5.5 m depth is the height difference between the crater rims and bottom, but it could mislead to be understood as water depth. What is the water depth? And did you drill through the ponding water (ice in May, right?) into the pingo core? P4, L3: Showing photos of cores will provide necessary information to judge stratigraphic integrity and possible contaminations. P4, L4: "Drill diameters", are these borehole's or core's diameters? What is the upper parts thickness? P4, L15-: Please provide information about subsampling interval for each measurement. P4, L27: This sentence indicates water was extracted from sample cores, but the following sentence obviously tells the cores were dried first, then added DI water. Probably, the water extraction is after the drying and DI water adding procedure? Please clarify. P5, L24: It is important to know how transparent and perfectly free from any inclusions (materials and bubbles) to understand formation of the massive ice. Could you show close-up photos of the ice? The next sentence mentions about 10P5, L25: Is the dimension 1-2 -10-20mm thickness or length of the flakes? What is 0.5 P5, L26: most air bubble should be rounded. Could you provide more information about the bubble shape? Oriented? Trained? Spherical other shape? What is 10 P5, L27: "well defined lower contact to the basal deposits", Please provide photo and describe more about characteristics of the boundary ice and sediment. P5, L27-28: "From 22.2 . . . is underlain by dark. . ." Ambiguous sentence. The layer 22.2-25m is the dark grey clay? Or within this layer the pingo ice is underlain by clay? P5, L29: Clear ice doesn't necessarily mean segregation ice. The word "clear" is vague in this case. Do you mean just color, bubble-free or no inclusions? P6, L1: "top of the pingo" could indicate entire pingo-top crater. Do you mean top of the crater rim? Highest point? P6, L2: What is "modern top soil" and "buried soil formation"? How did you differentiate them? "plant organic material" is terrestrial? P6, L6: 1.2m (4.7 -5.9m) thick clear segregation ice? This is interesting data to understand the formation of this kind of frost mounds. Including photos of these cores is really helpful also for relevant researchers to understand development mechanism. P6, L11: "structure less cryostructure" can be displayed by photo. P7, L3: "sedimentary water extracts" indicates you measured extract water from the original samples. See my comment on P4, L27. P7, 23-25: Why this points to the non-marine origin. Could you explain this in detail? This relates to one of your important conclusions. P9, L2: "on the mounts of valley sides..."? P9, L10: Is the source water exclusively from sub-glacial melt? Is there any contribution from rain or snow melt? P9, L19-20: I could not understand the logic of this sentence and the previous explanation. P9, L21: Do you have any information of discussion about how stable or variant geochemistry of the spring water seasonally and inter-annually? P10, L3-5: As you discussed earlier, the source water came from glacier melt in the upper area of the valley. Past precipitation (rain and snow) in the ground water source area would also be a probable water source for the pingo ice? "precipitation" in this paper should be well-defined because there are many types depending on time-scale, season, and precipitated area (if you discuss about groundwater source, precipitation in the recharge area might be different from that in your sampling area of precipitation and river data.). P11, 10-11: "Generally..." I cannot understand this sentence. Please revise it and provide some references. P11, L22-24: This is a long sentence and it's hard to understand. It is not clear to me why authors need to state this. It is better to bring comparable observations of internal structure of other pingos. P11, L28-30: Hard to understand. Please explain about "step-wise massive ice growth". P11, L33: I could not understand this logic. Do you mean the massive ice started to grow when freezing front reached at 15m depth and most of the upper sediment layer was lost by solifluction? P12, L10-12: I don't find the first half of this sentence is general fact. The second half seems to be too obvious to state. P12, L12-13: Please explain why Yoshikawa and Harada (1995) concluded this. P12, L16: Is the presence of gravelly sand and loam the only reason for non-marine deposit? It is unclear if authors suggestions are contradicting to marine deposits only or pingo growth after sea regression as referred by Yoshikawa and Harada (1995). P12, L18: This sentence doesn't sound to me because heaving amount that forms current pingo height could advance any moment of freezing

of intruding water. Are you assuming intrusion of groundwater into the pingo bottom started when the ground was frozen down to 15m? Is there any possibility the intrusion happened earlier, and advance of entire ground freezing and intrusive ice core development happen at the same time? P12, L34: Please explain and define "warm-based" and "cold-based" glaciers. P13, L2: What is "rare locations"?

P13, L10: active layer depth → active layer thickness, or in this case maximum thaw depth would be suitable. P13, L11: "fast" degradation → temporal degradation rate are not discussed in this paper. Authors should provide evidence of ongoing fast degradation and strong solifluction (against degraded in the past but stabilized) if they want to conclude this. P13, L18: It was hard to understand this sentence. Characteristics of what? Non-marine character of the pingo deposits indicate fault-related groundwater discharge and ground-water origin from warm-based glaciers are unlikely? But I couldn't understand why. P13, L22-24: This sentence needs to be rephrased or edited.

Table 1: What the "v" indicates? Fig 1: Provide contour lines information. Fig 1: Could you add information of geological faults location/direction? Fig 2: Why the sample intervals of stable isotopes and ions are so different, especially in the unit II? Fig 4 (a): One more value should be in the y-axis. Fig 2-5: Use different symbols for different components in same figures so that readers can distinguish them in blackwhite printouts. Fig 6: Please revise this image so that it can display difference between authors' and Yoshikawa Harada (1995) development models.

Please also note the supplement to this comment:
https://www.the-cryosphere-discuss.net/tc-2019-76/tc-2019-76-RC2-supplement.pdf

---

## Author Comment (AC1) · 31 Aug 2019

Reviewer 1 Interactive comment on "Pingo development in Grøndalen, West Spitsbergen" by Nikita Demidov et al. Trevor Porter (Referee)

The authors present stratigraphic profiles of solutes and water isotopes from pore ice of permafrost cores collected from the Fili pingo in West Spitsbergen, and of modern precipitation, tributaries, and a local spring to better constrain the origin of water in the pingo system. Based on the data, the authors deduce that the pingo is springfed, and that its evolution was characterised by several distinct periods of closed and semi-closed conditions, as evidenced by trends in the water isotope and solute data. They also use a Rayleigh isotope distillation model to show that the data diverge from a

closed-system. I found the methods and interpretation were robust. The paper was well written, easy to follow and the topic is well suited for The Cryosphere. Below are several comments meant to help the authors improve the communication of their work, and address one uncertainty that is not discussed. Following these minor revisions I would recommend this paper for publication. Answer: Thank you for your time and effort to review our manuscript. We appreciate your suggestions and answer them one-by-one. According changes in the manuscript are included in the revised version and referred in our replies.

Major comments. I am intrigued by the vertical trends in the water isotope dataset. To a large extent I agree with the interpretation assuming the following conditions: core #9 was drilled exactly in the centroid of the ice body; pingo geometry is conical; and it is reasonable to assume the pingo grew equally on all sides. Largely these points are not discussed. Isotopic stratigraphy of pingo ice could show 'apparent' reversals if the coring angle was off-axis, or pingo growth was asymmetric. Perhaps the authors can comment on this uncertainty. Again, I am in agreement with the interpretation, but would appreciate if this issue of coring angle and pingo growth geometry could be discussed. Answer: We agree that the interpretation of the vertical trends in the down-core profile strictly depends on the drilling position in the centroid of the massive ice body. Although the exact underground geometry of the massive ice has not been detected, we assume from the central drilling position at surface and concentric vertical drilling that the isotopic stratigraphy indeed represents the subsequent freezing stages of the pingo ice. Accordingly, we added in section 3.1 the following sentence: "The drilling position on top of the pingo was chosen in its center to assure that the centroid of the pingo ice body was captured in the core. The coring angle was held vertical." We further added the following sentence to section 5.2: "Assuming a conical geometry of the pingo ice body that grew equally to all sides, the chosen central drilling position on top of the pingo and the strictly vertical drilling allowed capturing subsequent freezing stages of the massive ice."

Minor comments. Abstract. The final 2 sentences are largely unconnected to the research. Please finish the abstract with some kind of significance statement instead. Answer: We changed the sentences as follows: "The presence of permafrost below the pingo ice body suggests that the talik is frozen and the water supply and pingo growth are terminated. The maximum thaw depth of the active layer reaching the top of the massive ice leads to its successive melt with crater development and makes the pingo extremely sensitive to further warming."

P2, L20-22. this sentence is too wordy, and confuses the message. please make it more concise. Answer: We changed the sentence as follows: "They differentiate into group I pingos fed by sub-permafrost groundwater along geologic faults, group II pingos fed by artesian flow of migrating sub-glacial groundwater mainly in river valley positions (in sensu Liestøl, 1977) and group III pingos (in sensu Yoshikawa and Harada, 1995). The latter are found in nearshore environments of post-glacial isostatic uplift and fed through small-scale discontinuities 'groundwater dikes' or taliks in aggrading permafrost within in marine deposits (Yoshikawa and Harada, 1995)."

P2, L24-26. please elaborate on why this is true. Answer: The statement from Liestøl (1996) is still true and based on the scarcity of data from the inner structure of pingos due to lacking drilling or exposures except for the studies by referred in the manuscript in section 1. We added the following elaboration: "This is still valid due to the scarcity of data from the inner structure of pingos because of rarely undertaken drilling."

P3, L26-27. the description of core thickness in relation to different base level elevations is a bit confusing. Please clarify in simple terms. Answer: We changed the sentence accordingly as follows: "The drilling of the Fili pingo in May 2017 started from the surface of central crater at 52.5 m asl and reached a depth of 11.5 m bs (core #9, 77.99355 °N, 14.66211 °E). The borehole was conserved and in April-May 2018 the drilling was continued in the same borehole down to a depth of 25 m bs"

P6, L17. The reported dD-d18O and d-dD slopes (6.7 and -0.2, respectively) are nearly

identical to the slopes observed in modern precipitation. Fig. 3 indicates dD-d18O slope is 6.78, and based on the dD-d18O equation the d-dD slope can be calculated to -0.18 (or -0.2 if rounded to 1 decimal). My point is, your claim that the effects of freezing on the co-isotope slopes are not well supported by the data, since precipitation has these slopes. Answer: To show difference in the d-$\delta$D data of the pingo massive ice (Figure 3c in the manuscript) and those of precipitation, we added here Figure R-1. Here is becomes obvious that the d-$\delta$D slope of precipitation is −0.06 and thus differs from those of the massive ice data. We therefore assume, that the co-isotope slopes of the massive ice as shown Figure 3 of the manuscript display the freezing effects during formation of the massive ice. The rather uncommon isotopic composition of modern precipitation are subject to a recent study by Skakun et al. (in review) where short-term variations in air mass trajectories are discussed to explain extrema in deuterium excess values. Taking further into account the large scatter in precipitation amounting to about 18‰ in $\delta$18O, to about 127 ‰ in $\delta$D (see Figure 3a and Table 1 in the manuscript) and to about 70 ‰ in d (see Figure R-1 above and Table 1 in the manuscript) if compared to those of the massive pingo ice the latter are distinctly smaller. Thus, if precipitation had been a major source for the pingo ice we would expect a much larger scatter in the isotopic composition. Reference: Skakun et al.: Stable isotopic content of atmospheric precipitation and natural waters in the vicinity of Barentsburg (Svalbard), Ice and Snow (ДžёДťЂ Đÿ ĐąĐ¡ĐţĐş), in review.

P9, L19-20. it is unclear how the previous sentence justifies this conclusion. Please elaborate. Answer: Based on literature data we assume a fast growth of the pingo massive ice. Accordingly, we changed the text as follows: "Estimations of pingo growth rate in Siberia and North America may reach values of order decimetres per year (Mackay, 1979; Chizhova and Vasil'chuk, 2018). Assuming a similar fast growth of the Fili pingo no or only little changes in isotopic composition of water source over the rather short period of pingo formation are likely. Thus, we assume the second controls on isotopic composition of the Fili pingo massive ice of less importance."

P10, L12-13. If true, you may be able to calculate the rate of pore water recharge based on deviation from isotope distillation model. Answer: The applied isotopic fractionation model does not allow calculating the admixture of water based on the deviation from the freezing line if the original isotopic composition of this additional source is unknown. There are two independent variables, and to find one, one needs to know the other, i.e., this problem is unsolvable.

P12, L10. . . .. valley evolution can already 'be' drawn. Answer: Changed accordingly.

Figures. The font size and resolution of some of the figures is too low for publication, and in some cases it was difficult to interpret the figures as given. Please revise to conform to the publication standards of The Cryosphere. Answer: To be changed accordingly in the final revision.

Please also note the supplement to this comment:
https://www.the-cryosphere-discuss.net/tc-2019-76/tc-2019-76-AC1-supplement.pdf

Figure R-1: Co-isotopic plot of *d* and δD in modern precipitation in
Barentsburg not included in the paper.

**Fig. 1.**

---

## Author Comment (AC2) · 31 Aug 2019

Reviewer 2 Interactive comment on "Pingo development in Grøndalen, West Spitsbergen" by Nikita Demidov et al. Go Iwahana (Referee)

The paper "Pingo development in Grondalen, West Spitsbergen" by Demidov et al. presents very rare data set about internal structure and ice geochemistry of a pingo in West Spitsbergen. The paper discusses possible water sources and freezing conditions of the core ice, then growth history the pingos in target area. Especially, the cryolithological information with geochemistry of entire pingo core ice (with underlain sediment) is a paramount value for understanding frozen ground on Earth and other planets. This paper should be published ultimately with additional information after

some clarifications of information provided and revision of discussion. Answer: We are grateful for your time and effort to thoroughly review our manuscript. We appreciate your suggestions and answer them one-by-one. According changes in the manuscript are included in the revised version and referred in our replies.

Major comments The title of paper is too broad, and it indicates overall study on pingo distribution and development history in Grondalen. However, the focus of this paper, to me, is unveiled internal structure of this particular pingo and interpretation of geochemistry of pingo ice and surrounding water. Could you revise the title so that reader can easily understand the contents of the paper more specifically? Answer: We changed the title accordingly to "Geochemical signatures of pingo ice and its origin in Grøndalen, West Spitsbergen".

Discussion about water source of pingo ice is not clear mainly because the definition of precipitation is vague in the text. I encourage authors to reconstruct the discussion considering time and area of the precipitation and groundwater. I think the confusion came from the fact that the water sources of pingo ice and sources of groundwater (and river) are different concepts (groundwater can be a source of pingo ice, but the groundwater itself has its water sources.). Answer: We differentiate generally three possible sources of water for the pingo ice: (1) atmospheric precipitation and its derivate as surface water, (2) sea water and (3) groundwater. Atmospheric waters feed the latter but due to subsurface turnover and interaction with rocks groundwater acquires geochemical signatures different from the atmospheric moisture. All three source have a sharply different composition from each other. The definition of precipitation in the manuscript solely refers to scarce modern precipitation stable isotope composition from 2016-17. Being limited to this we struggle to speculate on past precipitation, its past seasonality, precipitated area and groundwater recharge. In the course of our discussion we argue that atmospheric precipitation and surface waters are unlikely to be water sources that fed the pingo massive ice (see sections 5.1 and 5.2). Further elaborations on this topic are given in our replies to ref#2's specific minor comments below.

Section 5.3 should be rewritten and revise thoroughly to clearly present authors' discussion. Its paragraph structure does not match discussion flows. It was very hard to follow the logic of authors' idea and some statements don't sound to me as pointed in minor comments below. Authors discussed hydrologic conditions and history of pingo growth comparing to Yoshikawa and Harada (1995) model; however, explanation, evidence and reasoning to suggestion of non-marine sedimentation are weak and discrepancy points between researches are unclear. This can be improved by describing more details about Yoshikawa and Harada (1995)'s development model and their reasoning if you intend to include this comparison in the conclusion. Please make it clear about discrepancies and discussion in occurrence of sea regression at the target pingo location, timing of the regression, interpretation of sedimentation history at the site, and judgment of marine or non-marine sediment. Answer: In this paper we do not touch the age of ice and deposits. We studied the geochemical signatures of the pingo ice and discussed based on this the possible water sources of the pingo ice. In the discussion section, we allowed ourselves to make cautious assumptions about the sequence of events in the formation of the pingo. The upper sequences of deposits drilled by well 11 and well 10 are probably processed by the river and spread marine deposits. River processing of sediments is visible on space images - meandering of the channel along the whole length of Grøndalen can be clearly traced. Desalinization is visible from the results of the water extraction analysis.

I think it is important to show photo of obtained cores for judgment of integrity, and also to capture cryolithologic properties of the target pingo as authors indicated as one of the purposes. Recovery of the entire massive ice core of a pingo is a dominant value of this study, however, cryolithological description and discussion of the obtained core is poor. Aim (1) can be more developed by comparing to other pingos on Spitsbergen (and in other regions?). Answer: To our knowledge there are no comparable other records of internal pingo structures from Spitsbergen published. The Riverbed pingo in Adventdalen as referred in the manuscript (Yoshikawa, 1993) unfortunately was studied for other purposes, and it exposure in a distal position doesn't allow to comparison with

the Fili pingo in Grøndalen. We agree with importance of photos of obtained cores. Unfortunately the quality of photos is pure, so we add here Figure R-2, but suggest not to add this figure to manuscript and to leave alone core description in text form.

Specific minor comments P2, first paragraph: please provide some references for these descriptions. Answer: We agree with this comment and added the following reference to the manuscript: van Everdingen, R.E.: Multi‐language glossary of permafrost and related ground‐ice terms (revised 2005), Boulder, USA: National Snow and Ice Data Center/World Data Center for Glaciology, 1998.

P2, L25: What are the unresolved questions relevant to this paper? Answer: Following the recommendation from ref#1 we added the following sentence: "This is still valid due to the scarcity of data from the inner structure of pingos because of rarely undertaken drilling." The aims of the study follow in detail right afterward at the end of section 1.

P3, L23: I assume 5.5 m depth is the height difference between the crater rims and bottom, but it could mislead to be understood as water depth. What is the water depth? And did you drill through the ponding water (ice in May, right?) into the pingo core? Answer: Following the recommendation from ref#2 we added the following sentences: "The maximum water depth of the lake was >1 m. At the point of drilling, the ice thickness was 0.15 m."

P4, L3: Showing photos of cores will provide necessary information to judge stratigraphic integrity and possible contaminations. Answer: See Figure R-2. We would leave it to the editor whether to include such photographs into the Supplementary Material of the paper or not.

P4, L4: "Drill diameters", are these borehole's or core's diameters? What is the upper parts thickness? Answer: We changed the sentences accordingly to: "Core barrels outer diameters were 112 mm for the upper parts and 76 mm for the lower ones. The barrels wall thickness is 3 mm."

P4, L15-: Please provide information about subsampling interval for each measurement. Answer: We added the sentence: "The ice and permafrost deposits were sampled at intervals of about one half to one meter."

P4, L27: This sentence indicates water was extracted from sample cores, but the following sentence obviously tells the cores were dried first, then added DI water. Probably, the water extraction is after the drying and DI water adding procedure? Please clarify. Answer: We agree with this comment and changed the text accordingly as follows: "Sedimentary permafrost samples of cores #9, #10 and #11 were dried and sieved at 1 mm at the analytical laboratory of RAE-S, Barentsburg. Afterwards about 20 g of the dry sediment were suspended in 100 ml de-ionised water and filtered through 0.45 $\mu$m nylon mesh within 3 minutes after stirring to estimate the ion content after water extraction."

P5, L24: It is important to know how transparent and perfectly free from any inclusions (materials and bubbles) to understand formation of the massive ice. Could you show close-up photos of the ice? The next sentence mentions about 10% bubble inclusion in "single ice layers." What does this mean (I guess it is a volume, though)? In my experience, even very bubble-rich ground ice contained are space about 5% of total ice volume. Answer: We changed % to V%. See Figure R-2.

P5, L25: Is the dimension 1-2 -10-20mm thickness or length of the flakes? What is 0.5 %? Answer: We added 1-2 to 10-20 mm long. We changed % to V%.

P5, L26: most air bubble should be rounded. Could you provide more information about the bubble shape? Oriented? Trained? Spherical other shape? What is 10 %? Again, photo is the best way to display this information. Answer: We added following sentences. "In this particular layer and in all other layers of pingo massive ice bubbles are spherical and chaotic distributed. Most common are bubbles with diameter near 1 mm but some bubbles rich diameter up to 5 mm." See also Figure R-2.

P5, L27: "well defined lower contact to the basal deposits", please provide photo and

describe more about characteristics of the boundary ice and sediment. Answer: We changed paragraph as follow: "The 25 m long core #9 drilled from the pingo top crater exposed cover and basal sedimentary horizons enclosing massive pingo ice. From 0 to 1.5 m bs gravelly loam was found, which is assumed origin from the pingo top and moved downslope by cryoturbation and solifluction. Below this redeposited cover layer from 1.5 to 12 m bs transparent massive ice without any inclusions is observed. Air bubble content reaches up to 10 V% in single ice layers. In this particular layer and in all other layers of pingo massive ice bubbles are spherical and chaotic distributed. Most common are bubbles with diameter near 1 mm but some bubbles rich diameter up to 5 mm. Between 12 and 22.2 m bs the pingo ice remains transparent, but contains layers with 1-2 to 10-20 mm long large dark silty flakes in subvertical orientation (up to 0.5 V%). Alternating layers include rounded air bubbles (up to 10 %). The total thickness of the massive pingo ice amounts to 20.7 m. Its lower contact to the basal deposits is well defined in the core. Massive pingo ice near the contact was not rich in air bubbles and had small admixture of previously mentioned dark silty flakes. The lower end of the massive pingo ice in our core #9 was found at depth of 22.2 m bs. Below down to a depth of 25 m bs dark clay with regular reticulate and irregular reticulate cryostructures (ice lenses 2 to 20 mm thick) was found in the core. At 23.8-24.3 m bs ice lenses were absent but 2-4 mm long lenses of black clay material were present. At 22.3-23.5 m bs and at 23.7-23.8 m bs layers of transparent ice without inclusions and without air bubbles were found." See also Figure R-2.

P5, L27-28: "From 22.2 ... is underlain by dark..." Ambiguous sentence. The layer 22.2-25m is the dark grey clay? Or within this layer the pingo ice is underlain by clay? Answer: We changed sentences as follow. "The lower end of the massive pingo ice in our core #9 was found at depth of 22.2 m bs. Below down to a depth of 25 m bs dark clay with regular reticulate and irregular reticulate cryostructures (ice lenses 2 to 20 mm thick) was found in the core."

P5, L29: Clear ice doesn't necessarily mean segregation ice. The word "clear" is

vague in this case. Do you mean just color, bubble-free or no inclusions? Answer: We agree with this comment and specified the text accordingly as follows: "transparent ice without inclusions and without air bubbles".

P6, L1: "top of the pingo" could indicate entire pingo-top crater. Do you mean top of the crater rim? Highest point? Answer: We changed "top of the pingo" to "top of crater rim".

P6, L2: What is "modern top soil" and "buried soil formation"? How did you differentiate them? "plant organic material" is terrestrial? Answer: Modern soil - from the surface with living shrubs. The buried soil contains similar decomposed remains of vegetation. We changed sentence to: "The uppermost part from 0 to 2.5 m bs includes the modern top soil at 0 to 0.1 m bs with living shrub material and a buried soil formation at 0.25 to 0.4 m bs with decomposed similar shrub material."

P6, L6: 1.2m (4.7 -5.9m) thick clear segregation ice? This is interesting data to understand the formation of this kind of frost mounds. Including photos of these cores is really helpful also for relevant researchers to understand development mechanism. Answer: Please, see Figure R-2. Also taking in to account previous comment about "segregation ice" we changed the text as following: "From 2.5 to 12 m bs, the clay shows subhorizontal lenticular cryostructures up to 2 cm thick and includes ice-oversaturated deposits and ice with admixture of clay at 4.7- 5.9 m bs, at 6.65-7.05 m bs and at 8.2-8.6 m bs although the massive ice of the pingo was not reached. This ice and ice oversaturated deposits contain also sporadic gravel particles. In the layer 8.2-8.6 m ice contained up to 10 V% of spherical air bubbles with diameter near to 1 mm."

P6, L11: "structure less cryostructure" can be displayed by photo. Answer: See Figure R-2.

P7, L3: "sedimentary water extracts" indicates you measured extract water from the original samples. See my comment on P4, L27. Answer: We changed the according method description in section 3.4 to make clear what sedimentary water extracts stands

for as follows: "Sedimentary permafrost samples of cores #9, #10 and #11 were dried and sieved at 1 mm at the analytical laboratory of RAE-S, Barentsburg. Afterwards about 20 g of the dry sediment were suspended in 100 ml de-ionised water and filtered through 0.45 $\mu$m nylon mesh within 3 minutes after stirring to estimate the ion content after water extraction."

P7, 23-25: Why this points to the non-marine origin. Could you explain this in detail? This relates to one of your important conclusions. Answer: The section 4.4.2 solely presents the results that are further discussed in discussion section 5. Therefore, no interpretation of the data is given here. We deleted last part of the sentence "pointing to the non-marine origin of the deposits" from the text.

P9, L2: "on the mounts of valley sides..."? Answer: We agree with this comment and changed the term accordingly to: "Glaciers on the mounts surrounding the valley".

P9, L10: Is the source water exclusively from sub-glacial melt? Is there any contribution from rain or snow melt? Answer: This is discussed earlier in section 5.1 as follows: "Precipitation and surface waters in Grøndalen have lower ion contents and different composition if compared to the pingo massive ice (Table 2), which also excludes these sources as the main ones for the pingos of Grøndalen." And further in section 5.2 as follows: "This is much lower than the modern mean annual precipitation values in the Barentsburg region with $-9.0\pm4.2$ ‰ in $\delta$18O and $-64\pm30$ ‰ in $\delta$D, and closer to mean values of the Grøn River and its tributaries with about $-11.7\pm0.3$ ‰ in $\delta$18O and $-78\pm3$ ‰ in $\delta$D (Table 1). Thus, if precipitation was the main source of the massive ice, its onset took place during a colder period than today. More likely, the underground water sources feeding spring near the pingo was the same source for the massive pingo ice." We also point out that underground water is currently isolated from surface water due to permafrost.

P9, L19-20: I could not understand the logic of this sentence and the previous explanation. Answer: Based on literature data we assume a fast growth of the pingo massive

ice. Accordingly, we changed the text as follows: "Estimations of pingo growth rate in Siberia and North America may reach values of order decimetres per year (Mackay, 1979; Chizhova and Vasil'chuk, 2018). Assuming a similar fast growth of the Fili pingo no or only little changes in isotopic composition of water source over the rather short period of pingo formation are likely. Thus, we assume the second controls on isotopic composition of the Fili pingo massive ice of less importance."

P9, L21: Do you have any information of discussion about how stable or variant geochemistry of the spring water seasonally and inter-annually? Answer: Unfortunately, we have yet only single point observations. The only data available is included in our study. Therefore, any temporal variability of the spring water stable isotope composition remains unknown. We can only point out that in spring and summer 2019 we observed the water coming from this spring and that organoleptic property of water was the same.

P10, L3-5: As you discussed earlier, the source water came from glacier melt in the upper area of the valley. Past precipitation (rain and snow) in the ground water source area would also be a probable water source for the pingo ice? "precipitation" in this paper should be well-defined because there are many types depending on time-scale, season, and precipitated area (if you discuss about groundwater source, precipitation in the recharge area might be different from that in your sampling area of precipitation and river data.). Answer: As already stated above to reply on comment P9, L10, precipitation and surface waters are unlikely to be the water source feeding the growing pingo ice. This is given in section 5.1 as follows: "Precipitation and surface waters in Grøndalen have lower ion contents and different composition if compared to the pingo massive ice (Table 2), which also excludes these sources as the main ones for the pingos of Grøndalen." And further in section 5.2 as follows: "This is much lower than the modern mean annual precipitation values in the Barentsburg region with $-9.0\pm4.2$ ‰ in $\delta 18O$ and $-64\pm30$ ‰ in $\delta D$, and closer to mean values of the Grøn River and its tributaries with about $-11.7\pm0.3$ ‰ in $\delta 18O$ and $-78\pm3$ ‰ in $\delta D$ (Table 1). Thus,

if precipitation was the main source of the massive ice, its onset took place during a colder period than today. More likely, the underground water sources feeding spring near the pingo was the same source for the massive pingo ice." Please, find also our reply to the comment of ref#1 on precipitation data, P6, L17: To show difference in the d-$\delta$D data of the pingo massive ice (Figure 3c in the manuscript) and those of precipitation, we added here Figure R-1. Here is becomes obvious that the d-$\delta$D slope of precipitation is –0.06 and thus differs from those of the massive ice data. We therefore assume, that the co-isotope slopes of the massive ice as shown Figure 3 of the manuscript display the freezing effects during formation of the massive ice. The rather uncommon isotopic composition of modern precipitation are subject to a recent study by Skakun et al. (in review) where short-term variations in air mass trajectories are discussed to explain extrema in deuterium excess values.

Taking further into account the large scatter in precipitation amounting to about 18‰ in $\delta$18O, to about 127 ‰ in $\delta$D (see Figure 3a and Table 1 in the manuscript) and to about 70 ‰ in d (see Figure R-1 and Table 1 in the manuscript) if compared to those of the massive pingo ice the latter are distinctly smaller. Thus, if precipitation would have been a major source for the pingo ice we would expect a much larger scatter in the isotopic composition. Reference: Skakun et al.: Stable isotopic content of atmospheric precipitation and natural waters in the vicinity of Barentsburg (Svalbard) in 2016-2017, Ice and Snow (ĐŽёĐťˇ Đÿ ĐąĐ¡ĐţĐş), in review.

The only data available to us yet is modern precipitation stable isotope composition from 2016-17. Being limited to this we struggle to speculate on past precipitation, its seasonality and precipitated area.

P11, 10-11: "Generally. . ." I cannot understand this sentence. Please revise it and provide some references. Answer: We deleted accordingly this sentence and the previous one from the text.

P11, L22-24: This is a long sentence and it's hard to understand. It is not clear to

me why authors need to state this. It is better to bring comparable observations of internal structure of other pingos. Answer: To our knowledge there are no comparable other records of internal pingo structures from Spitsbergen published. The Riverbed pingo in Adventdalen as referred here unfortunately was studied for other purposes, and it exposure in a distal position doesn't allow to comparison with the Fili pingo in Grøndalen. Because pingos are to our understanding features mainly controlled by local hydrological and morphological conditions any comparison of the Fili pingo record to those from other arctic regions seems inappropriate. To clarify our thought we changed the sentence as following: The Riverbed pingo in Adventdalen as referred in (Matsuoka et al., 2004) unfortunately was studied for other purposes, and it exposure in a distal position doesn't allow to comparison with the Fili pingo in Grøndalen, where borehole in the centre of mound exhibits the entire massive ice.

P11, L28-30: Hard to understand. Please explain about "step-wise massive ice growth". Answer: We deleted accordingly 'stepwise' from this sentence.

P11, L33: I could not understand this logic. Do you mean the massive ice started to grow when freezing front reached at 15m depth and most of the upper sediment layer was lost by solifluction? Answer: The thickness of the massive ice (20.7 m) exceeds the height of the pingo of 9.5 m, which is explained by massive ice growth when the ongoing subsurface freezing pushed the previously formed ice and cover deposits upward. The latter moved subsequently from the pingo top down-slope by solifluction. Thus, the 9.5 m amplitude of surface uplift seen in the modern stage of pingo evolution became less than the 20.7 m thickness of the massive ice.

P12, L10-12: I don't find the first half of this sentence is general fact. The second half seems to be too obvious to state. Answer: We deleted accordingly this sentence.

P12, L12-13: Please explain why Yoshikawa and Harada (1995) concluded this. Answer: The original paper there is no explanation or more detailed elaboration given. The relevant section reads as follows: "Pingos in Grøndallen and at the mouth of the

Reindalen are situated about 50 m above sea level. This is the same level as the maximum elevation of Holocene marine deposits. The pingo overburden is composed of marine material. There are several generations of pingos that started growth quickly after retreat of the sea. They do not have non-marine sediments over the marine deposits as does the area surrounding the pingos. These pingos are considered former group III pingos."

P12, L16: Is the presence of gravelly sand and loam the only reason for non-marine deposit? It is unclear if authors suggestions are contradicting to marine deposits only or pingo growth after sea regression as referred by Yoshikawa and Harada (1995). Answer: The statement of Yoshikawa and Harada (1995) is shown above. To our understanding this statements claims pingo growth within marine deposits as characteristic for group III pingos which is unfortunately not based on specific data from Grøndalen in the paper by Yoshikawa and Harada (1995). Besides the granulometric properties of the Grøndalen valley deposits also the sedimentary water extract hydrochemistry from our study as described in sections 4.4.2 and 4.4.3 and discussed in section 5.1 clearly show the terrestrial (non-marine, or reworked marine) origin of the valley deposits of Grøndalen in which the pingos formed. Thus, our suggestions do not contradict pingo growth after deglaciation and sea level regression, but the marine character of the deposits in which the Grøndalen pingos formed.

P12, L18: This sentence doesn't sound to me because heaving amount that forms current pingo height could advance any moment of freezing of intruding water. Are you assuming intrusion of groundwater into the pingo bottom started when the ground was frozen down to 15m? Is there any possibility the intrusion happened earlier, and advance of entire ground freezing and intrusive ice core development happen at the same time? Answer: Here we rely on the conventional mechanism of formation of swelling hillock by means of moisture migration to the freezing front (Kudryavtsev V.A. (ed.): Obshcheye Merzlotovedeniye (Geocryology). Moscow, Izdatelstvo Moskovskogo Universiteta, 1978 (in Russian)). According to this mechanism, the lower boundary

of the ice core will correspond to the position of the freezing front at which the ice formation began.

P12, L34: Please explain and define "warm-based" and "cold-based" glaciers. Answer: We use this commonly used in glaciology terms. For example in (Bennet M.R. and Glasser N.F.: Glacial geology: ice sheets and landforms. Oxford, Wiley-Blackwell. 385 pp.,2009) they are explained as following. Cold-based glaciers are glaciers which are frozen to their beds and no meltwater is present at the ice-bed interface. In contrast, in warm-based glaciers basal ice is constantly melting and the ice-bed interface is therefore lubricated with meltwater.

P13, L2: What is "rare locations"? Answer: We agree with this comment and deleted accordingly "at rare locations" from this sentence.

P13, L10: active layer depth -> active layer thickness, or in this case maximum thaw depth would be suitable. Answer: We agree with this comment and changed the term accordingly to: "maximum thaw depth".

P13, L11: "fast" degradation -> temporal degradation rate are not discussed in this paper. Authors should provide evidence of ongoing fast degradation and strong solifluction (against degraded in the past but stabilized) if they want to conclude this. Answer: In our text it is written "The maximum thaw depth of 1.5 m in September 2018 reached the uppermost massive ice, which indicates the ongoing fast degradation of the pingo. This is further seen in the crater lake on top of the pingo and strong solifluction that removes cover deposits downslope." It means that from temperature measurements in the borehole we know that zero degrees isotherm reaches upper boundary off massive ice in the end of warm season leading to irrevocable melting of ice and crater deepening. In addition to temperature measurements it must be mentioned that upper part of 3 m long plastic drive pipe which had been inserted to the borehole to prevent water propagation to the borehole from active later after end of first stage of drilling in May 2017 was found in April 2018 declinated on 0.5 m in the direction of solifluction

removal.

P13, L18: It was hard to understand this sentence. Characteristics of what? Non-marine character of the pingo deposits indicate fault-related groundwater discharge and ground-water origin from warm-based glaciers are unlikely? But I couldn't understand why. Answer: Changed accordingly to: "In the Fili pingo record of Grøndalen we concurrently identified pingo-formation characteristics such as fault-related ground-water discharge (typical for group I pingos) and ground-water origin from warm-based glaciers (typical for group II pingos). The proposed pingo formation in Grøndalen is connected to epigenetic refreezing of marine deposits (typical for group III pingos) but not immediately after sea regression due to the reworking of marine sediments (or nonmarine origin of sediments) seen in deposits surrounding the Grøndalen pingos."

P13, L22-24: This sentence needs to be rephrased or edited. Answer: Changed accordingly to: "The origin and distribution of pingos in Grøndalen depends on the complex interaction of hydrogeological conditions and sea level, glaciers and permafrost dynamics superimposed by climate variability over time. The latter may be typical for vast archipelago and makes investigation of pingos important for understanding key stages of cryosphere evolution of Spitsbergen."

Table 1: What the "v" indicates? Answer: The "v" in two places in Table 1 is an artefact from previous draft versions of the manuscript. It is replaced by the minus symbol "—" in the revised manuscript.

Fig 1: Provide contour lines information. Answer: Contour lines information is shown in revised figure.

Fig 1: Could you add information of geological faults location/direction? Answer: Geological faults are shown in the revised figure.

Fig 2: Why the sample intervals of stable isotopes and ions are so different, especially in the unit II? Answer: We tried to take samples every 0.5 m. The irregularity of the

samples analysed is due to the loss of samples during drilling, storage and analysis.

Fig 4 (a): One more value should be in the y-axis. Answer: Changed accordingly in the revised Figure 4.

Fig 2-5: Use different symbols for different components in same figures so that readers can distinguish them in black&white printouts. Answer: For clarity of the figures and to better distinguish single data points and curves, we'd prefer to present our data in colour plots. We'd leave the decision with the handling editor and the production manager of "The Cryophere" whether to show black-and-white compatible or colour figures in the final version.

Fig 6: Please revise this image so that it can display difference between authors' and Yoshikawa & Harada (1995) development models. Answer: Our figure reflects the specific situation in the Grøndalen Valley. Here, the retreat of the sea, the onset of freezing, and the migration of moisture from the bottom melting of glaciers through the underground aquifer appear, which was not mentioned in the article by Yoshikawa & Harada (1995).

Please also note the supplement to this comment:
https://www.the-cryosphere-discuss.net/tc-2019-76/tc-2019-76-AC2-supplement.pdf

Figure R-1: Co-isotopic plot of *d* and δD in modern precipitation in
Barentsburg not included in the paper.

**Fig. 1.**

[Figure]

Figure R-2: Photos of obtained cores: A - gravelly loam at 0.25-0.4 m core #9 with wavy cryostructure, ice lenses up to 1 mm thick, B – transparent pingo ice containing dark silty flakes at 17.5-17.7 m core #9, C - crossection of pingo ice with dark silty flakes and air bubbles at 21.5-21.6 m core #9, D – boundary between massive ice and underlying sediments at 22.2 m core #9, E - dark clay with irregular reticulate cryostructures (ice lenses up to 5 mm thick) at 24.9-25.0 m core #9, F - ice-oversaturated deposits of core #10 at 4.95-5.2 m, G- gravelly sand with structureless cryostructure at 3.9-4.0 m core #11.

[Figure]

**Fig. 2.**

**Supplement:**

**Geochemical signatures of pingo ice and its origin in Grøndalen, West Spitsbergen**

Nikita Demidov[1], Sebastian Wetterich[2], Sergey Verkulich[1], Aleksey Ekaykin[1,3], Hanno Meyer[2], Mikhail Anisimov[1,3], Lutz Schirrmeister[2], Vasily Demidov[1], Andrew J. Hodson[4, 5]

[1]Arctic and Antarctic Research Institute, Bering St. 38, 199397 St. Petersburg, Russia

[2]Alfred Wegener Institute Helmholtz Center for Polar and Marine Research, Telegrafenberg A45, D-14473 Potsdam, Germany

[3]St. Petersburg State University, 10[th] Line 33-35, 199178 St. Petersburg, Russia

[4]University Centre in Svalbard, N-9171 Longyearbyen, Norway

[5]Western Norway University of Applied Sciences, Røyrgata 6, N-6856 Sogndal, Norway

*Correspondence to*: Nikita Demidov (nikdemidov@mail.ru)

**Abstract.** Pingos are common features in permafrost regions that form by subsurface massive-ice aggradation and create hill-like landforms. Pingos on Spitsbergen have been previously studied to explore their structure, formation timing, connection to springs as well as their role in post-glacial landform evolution. However, detailed hydrochemical and stable-isotope studies of massive ice samples recovered by drilling has yet to be used to study the origin and freezing conditions in pingos. Our core record of 20.7 m thick massive pingo ice from Grøndalen differentiates into four units: two characterised by decreasing $\delta^{18}O$ and $\delta D$ and increasing $d$ (units I and III), and two others show the opposite trend (units II and IV). These delineate changes between episodes of closed-system freezing with only slight recharge inversions of the water reservoir, and more complicated episodes of groundwater freezing under semi-closed conditions when the reservoir got recharged. The water source for pingo formation shows similarity to spring water data from the valley with prevalent $Na^+$ and $HCO_3^-$ ions. The sub-permafrost groundwater originates from subglacial meltwater that most probably followed the fault structures of Grøndalen and Bøhmdalen. The pPresence of permafrost underbelow the pingoice body of pingo suggests that the talik is frozen, and termination of the water supply and pingo growthare terminated. The mMaximum 
[revised manuscript text omitted]

---

## Author Response (AR3)

Dear Peter Morse,

Thank you for handling our submission and for your final edits on the revised manuscript. We re-formatted accordingly the Tables 1-3. The DOI registration of the entire dataset at PANGAEA is still pending. We hope to receive the data DOI these days and will inform you as soon as possible.

Comment: Manuscript looks good except for the tables. I have provided clearer direction, and attached an example of a properly formatted table.

The 3 tables are still inverted. The independent variable (the site or water source) that you change goes in the first column, and the dependent variables (the measurements or test results) follow in subsequent

10 columns.

"n" should become " Number of subsamples [n]"

Answer: Thank you very much. We changed the Tables 1-3 accordingly.

[revised manuscript text omitted]

---

## Author Response (AR4)

Dear Dr. Demidov,

Thank you for your edits. The tables are in good shape now.

The only problem that I see is with respect to Skakun et al. (in review) and the data for river and tributary water of the Grøn River. The methods section indicates that your team did all of the sampling, but the table descriptions indicate that there are additional data from Skakun et al. This needs to be made more transparent.

It would be best if your article is not contingent on another manuscript that is not published and in review. Manuscript preparation guidelines for authors indicate in the References section that "Works cited in a manuscript should be accepted for publication of published already." However, if there is no alternative, you can cite work that is "in review", but this is not ideal.

Please either make it more clear in the methods section that you have no alternative to using this data and add the appropriate citation, or contact Skakun et al. and make arrangements to publish the data here (they can then refer to your published work).

I regret that I did not catch this earlier, but there is time as you are waiting for the DOI to be assigned to your Original data.

Best regards,

Peter

Dear Peter Morse,

we apologize for our delayed reply due to travel in the last two weeks. Following your latest comments, we checked the recent publication stage of the Skakun et al. manuscript and are happy to report that this publication got accepted and is planned to be published in the first volume of Лёд и Снег in 2020. The according citation reads now:

Skakun, A.A., Chikhachev, K.B., Ekaykin, A.A., Kozachek, A.V., Vladimirova, D.O., Veres, A.N., Verkulich, S.R., Sidorova, O.R., Demidov, N.E.: Stable isotopic content of atmospheric precipitation and natural waters in the vicinity of Barentsburg (Svalbard) (Изотопный состав атмосферных осадков и природных вод в районе Баренцбурга (Шпицберген)), Ice and Snow (Лёд и Снег), 60(1), accepted, https://ice-snow.igras.ru/jour/issue/archive, 2020 (in Russian).

Because the Лёд и Снег journal (https://ice-snow.igras.ru/jour/issue/archive) does not provide 'in press' (online available) publications prior to final production, we unfortunately cannot provide any better information yet. However, we hope this reference as accepted paper is now in accordance with the guidelines of The Cryosphere. If there will be more information available such as page numbers and DOI before the final production of our ms, we will include those during the proofreading process latest.

We furthermore added the following sentence to section 3.3 Stable water isotopes: "Additional samples from surface waters (Grøn River and tributaries) and precipitation (2016-2017) were likewise analysed and are presented in detail by Skakun et al. (2020)." to make clear that precipitation and surface water data refer to the study by Skakun et al.

We updated the revised manuscript accordingly and further added the data DOI from the PANGAEA database for our original dataset.

Best regards, Nikita Demidov and Sebastian Wetterich

[revised manuscript text omitted]